# Tissue memory CD4+ T cells expressing IL-7 receptor-alpha (CD127) preferentially support latent HIV-1 infection

Feng Hsiao[1,2☯], Julie Frouard[1,2☯], Andrea Gramatica[1], Guorui Xie[1,2], Sushama Telwatte[3], Guinevere Q. Lee[4], Pavitra Roychoudhury[5], Roland Schwarzer[1], Xiaoyu Luo[1], Steven A. Yukl[3], Sulggi Lee[6], Rebecca Hoh[7], Steven G. Deeks[7], R. Brad Jones[4], Marielle Cavrois[1], Warner C. Greene[1,8], Nadia R. Roan[1,2]*

1 Gladstone Institute of Virology and Immunology, San Francisco, California, United States of America, 2 Department of Urology, University of California, San Francisco, California, United States of America, 3 San Francisco Veterans Affairs (VA) Medical Center and University of California, San Francisco, California, United States of America, 4 Department of Medicine, Division of Infectious Diseases, Weill Cornell Medicine, New York, New York, United States of America, 5 Department of Laboratory Medicine, University of Washington, Seattle, Washington, United States of America, 6 Zuckerberg San Francisco General Hospital and the University of California, San Francisco, California, United States of America, 7 Division of HIV, Infectious Diseases and Global Medicine, University of California San Francisco, San Francisco, California, United States of America, 8 Departments of Medicine and Departments of Microbiology and Immunology, University of California, San Francisco, California, United States of America

☯ These authors contributed equally to this work.
* nadia.roan@ucsf.edu

**Data Availability Statement:** All relevant data are within the manuscript and its Supporting Information files.

## Abstract

The primary reservoir for HIV is within memory CD4+ T cells residing within tissues, yet the features that make some of these cells more susceptible than others to infection by HIV is not well understood. Recent studies demonstrated that CCR5-tropic HIV-1 efficiently enters tissue-derived memory CD4+ T cells expressing CD127, the alpha chain of the IL7 receptor, but rarely completes the replication cycle. We now demonstrate that the inability of HIV to replicate in these CD127-expressing cells is not due to post-entry restriction by SAMHD1. Rather, relative to other memory T cell subsets, these cells are highly prone to undergoing latent infection with HIV, as revealed by the high levels of integrated HIV DNA in these cells. Host gene expression profiling revealed that CD127-expressing memory CD4+ T cells are phenotypically distinct from other tissue memory CD4+ T cells, and are defined by a quiescent state with diminished NFκB, NFAT, and Ox40 signaling. However, latently-infected CD127+ cells harbored unspliced HIV transcripts and stimulation of these cells with anti-CD3/CD28 reversed latency. These findings identify a novel subset of memory CD4+ T cells found in tissue and not in blood that are preferentially targeted for latent infection by HIV, and may serve as an important reservoir to target for HIV eradication efforts.

## Author summary

Although the primary targets of HIV are CD4+ T cells that reside in tissues, to date most studies characterizing HIV-infected cells have used cells isolated from blood. We recently

**Funding:** This work was supported by the National Institutes of Health (R01AI127219 and R01AI147777 to N.R.R.; P01AI131374 to N.R.R. and W.C.G.; R01DK108349, R01DK120387, and R01AI132128 to S.A.Y.) and the amfAR Institute for HIV Cure Research (109301). We also acknowledge NIH for the sorter (S10-RR028962) and support from CFAR (P30AI027763) and the Pendleton Foundation. The funders had no role in study design, data collection and analysis, decision to publish, or preparation of the manuscript.

**Competing interests:** The authors have declared that no competing interests exist.

demonstrated that although HIV can enter a diverse array of CD4+ T cells present in lymphoid tissues, only a limited subset of these cells support completion of the full viral life cycle. Here, we identified a subset of lymphoid tissue CD4+ T cells that express a surface protein called CD127. These cells efficiently support early stages of the viral life cycle up to the point where the virus has integrated its DNA into the human genome, but suppress the later stages of viral gene expression. Relative to other types of CD4+ T cells, the CD127+ cells exist in a more quiescent state, which may be inadequate for HIV protein expression and results in latency. This latency, however, can be reversed following T cell stimulation. Better characterizing the mechanisms that promote latency in tissue cells, using CD127 as a biomarker or this specific subset of cells as a model system, promises to help identify new approaches for eliminating or controlling these cells thus contributing to achieving a functional cure for HIV/AIDS.

## Introduction

Although HIV-1 is capable of entering a diverse array of CD4+ T cells, multiple post-entry blocks can limit completion of the full viral life cycle. Early blocks that occur soon after viral entry can prevent efficient reverse transcription and integration, and are among the major post-entry restriction mechanisms preventing HIV-1 from efficiently infecting resting CD4 + T cells from blood [1, 2]. This restriction has been largely attributed to the activity of SAMHD1, a phosphohydrolase that depletes the intracellular pool of dNTPs needed for reverse transcription [3, 4]. Multiple mechanisms also restrict the later stages of the viral life cycle, after the provirus has integrated into the host genome. For instance, mechanisms that prevent initiation or elongation of transcription from the viral LTR, splicing and export of HIV transcripts, and/or translation of HIV proteins can prevent completion of the viral life cycle, thereby promoting HIV latency [5, 6]. Given that latently-infected cells persist for the lifetime of an individual [7], a better understanding of the types of cells that harbor integrated provirus but restrict HIV gene expression can lead to important insights for approaches to eliminate these cells as a strategy for curing HIV-infected individuals.

To date, most studies comparing the susceptibility of different CD4+ T cells to infection by HIV have been conducted on cells isolated from human blood. However, the vast majority of CD4+ T cells reside in tissues, and these cells exhibit phenotypic features distinct from those in blood [8]. Further emphasizing the need to study the properties of cells from tissues, a recent study in a non-human primate model of HIV infection demonstrated that more than 99% of infected cells are found in lymphoid tissues, and nearly ~36% of these in lymph nodes [9].

We recently conducted a CyTOF study to better understand the types of tissue CD4+ T cells that are most and least susceptible to infection by HIV [10]. CyTOF is a mass spectrometry-based single-cell protein quantitation technique analogous to flow cytometry that employs antibodies conjugated to rare isotopes instead of fluorophores [11]. Because isotopes have discrete readouts and are not limited by spectral overlap, many more variables can be assessed with CyTOF than with conventional flow cytometry. By comparing cells capable of fusing to CCR5-tropic HIV through use of the virion fusion assay [12] to those which HIV was capable of productively infecting as revealed by expression of an LTR-driven reporter gene, we were able to determine which subsets harboring HIV virions went on to become productively infected. Our analysis revealed that CCR5-tropic HIV could not efficiently enter naïve CD4 + T cells, but could enter all subsets of memory CD4+ T cells [10]. Nevertheless, only limited

subsets of memory CD4+ T cells supported productive infection. Notably, memory CD4+ T cells expressing CD127, the alpha chain of the IL7 receptor, efficiently fused with HIV but did not allow the virus to express its reporter gene. In contrast, memory CD4+ T cells expressing CD57, a marker of terminally differentiated cells and T follicular helper (Tfh) cells, readily supported productive infection. Additional studies using purified CD127+ memory CD4+ T cells demonstrated that the lack of productive infection was not due to slower viral replication kinetics or to downregulation of cell-surface CD127 following HIV infection [10].

In this study, we set out to better characterize the molecular basis for the block in HIV replication in tissue CD127+ memory CD4+ T cells by considering two main possibilities. First, because SAMHD1 is preferentially expressed in cells residing within the extrafollicular region of tonsillar explants [3], where the CD127+ cells are located [10, 13], we tested the possibility that SAMHD1 was actively restricting viral infection in these cells. Having found that this was not the case, we then considered the possibility that the CD127+ cells preferentially undergo latent infection, particularly since IL7, the ligand for CD127, has been implicated in driving persistence of latently-infected cells *in vivo* via homeostatic proliferation [14]. We demonstrate that tonsillar memory CD4+ T cells expressing CD127 are indeed biased to undergo latent infection, and further characterize host features associated with suppression of viral gene expression in these cells.

## Results

### Tissue-derived memory CD4+ T cells expressing CD127 restrict productive infection by HIV-1

We previously demonstrated by CyTOF that tonsillar memory CD4+ T cells can be categorized into three mutually exclusive subsets: CD57+CD127- cells (hereafter referred to as CD57 + Tm cells), CD57-CD127+ cells (hereafter referred to as CD127+ Tm cells), and cells expressing neither CD57 nor CD127 (hereafter referred to as CD57-CD127- Tm cells). The CD127 + Tm subset efficiently fuses to HIV but does not support productive infection [10]. To verify this observation and to assess how generalizable these findings were, we repeated these experiments using tonsillar cells from a total of 15 different donors and analyzed the data by flow cytometry. Unstimulated human lymphocyte aggregate cultures (HLACs) from tonsils were mock-treated or exposed to F4.HSA, a CCR5-tropic HIV-1 that encodes the transmitted/founder envelope C.109FPB4 and expresses as a reporter heat-stable antigen (HSA) under the control of the HIV LTR [10]. Three days later, cells were harvested for analysis by flow cytometry. Consistent with the results from CyTOF, distinct populations of CD57+, CD127+, and CD57-CD127- Tm cells were readily detected among memory CD4+ T cells in the mock-treated sample; in striking contrast, the productively-infected (HSA+) cells were made up almost exclusively of only the CD57+ and CD57-CD127- Tm cell populations (Fig 1A). The low infection rates in the CD127+ Tm cells were not the result of a low frequency of these cells in HLACs, since infection rates in CD127+ Tm cells were very low even in donors that harbored high frequencies of these cells (S1 Fig). Quantitation of datasets from the 15 donors revealed that the proportion of infected CD127+ Tm cells was significantly lower (p<0.0001) than the proportion of uninfected CD127+ Tm cells (Fig 1B). In comparison, the CD57+ Tm cells were over-represented within productively infected cells (p<0.001) while the proportions of CD57-CD127- among the uninfected and infected cells were not significantly different (Fig 1B).

To determine whether the inability of F4.HSA to infect the CD127+ Tm population is a tissue-specific phenomenon, we conducted a similar experiment using unstimulated PBMCs as target cells. In contrast to HLACs, memory CD4+ T cells from uninfected PBMCs harbored a

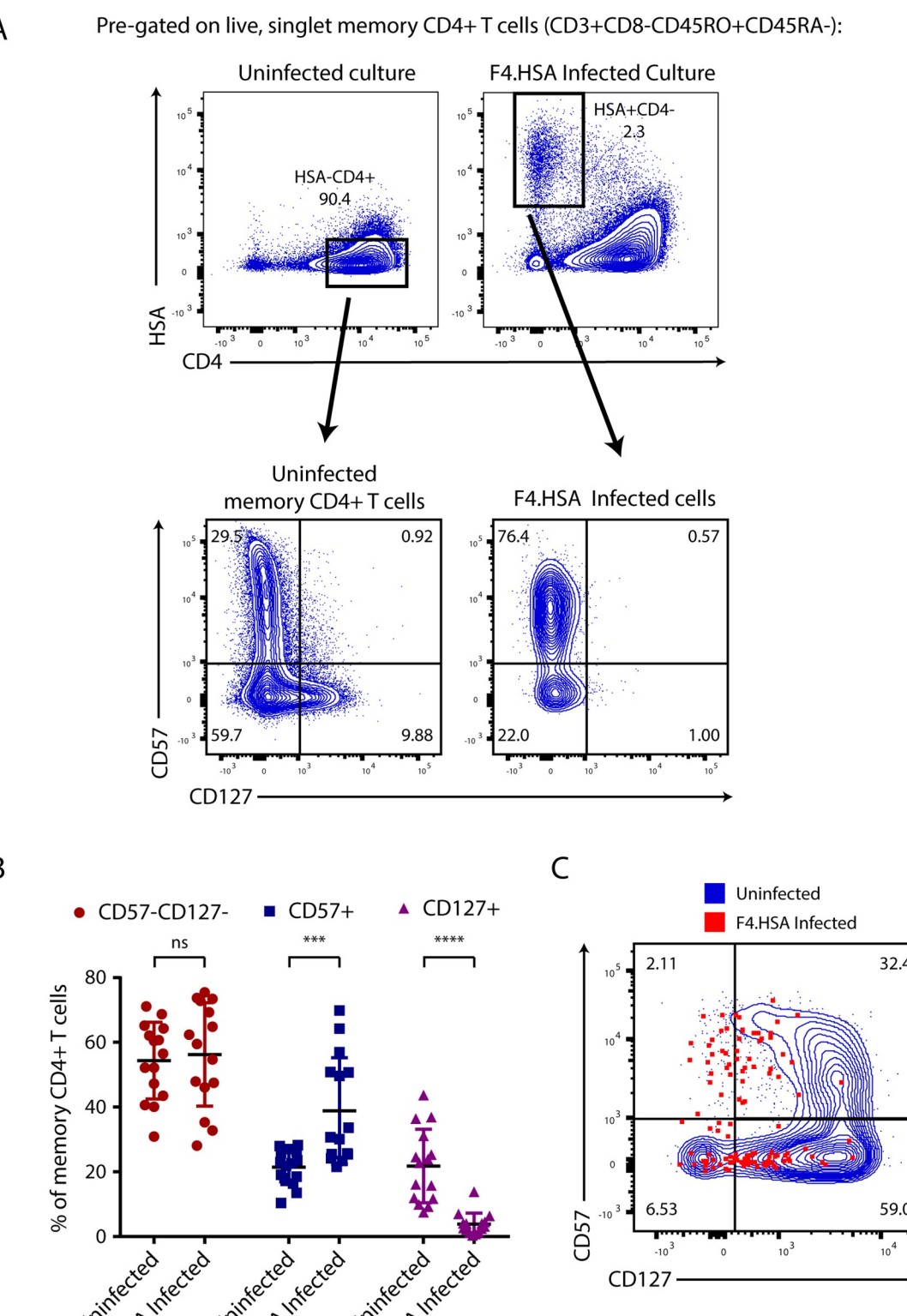

**Fig 1. CD127+ memory CD4+ T cells from tonsils are poorly susceptible to productive infection by HIV-1.** A) CD127+ Tm cells are preferentially absent amongst infected tonsillar cells. HLACs were mock-treated or exposed for 3 days to the CCR5-tropic reporter virus F4.HSA, after which the populations of uninfected memory CD4+ T cells (*top left*) and infected memory CD4+ T cells that have downregulated cell-surface CD4 (*top right*) were assessed for expression levels of CD57 and

CD127 (*bottom plots*). B) Bar graph comparing the proportions of CD57+, CD127+, and CD57-CD127- Tm cells amongst uninfected or HIV-infected Tm cells from 15 independent donors. ***p<0.001, ****p<0.0001 as determined using a 2-tailed paired parametric t-test. C) CD127+ Tm cells from blood do not restrict productive infection by F4.HSA. Infection conditions were set up as described in panel A but using PBMCs instead of HLACs. Memory CD4+ T cells from the uninfected culture are shown in blue while productively-infected (HSA+) memory CD4+ T cells are shown in red. All results in this figure are pre-gated on live, singlet CD3+CD8-CD45RO+CD45RA- cells.

large population of cells expressing high levels of both CD57 and CD127. Infected cells were present among the CD57+CD127+ cells, as well as the CD127+ Tm cells (Fig 1C). Therefore, the inability of F4.HSA to infect CD127+ Tm cells seems restricted to tissue cells.

Since T cell activation is known to partially relieve some post-entry restriction mechanisms [3], we next assessed whether prior stimulation of HLACs with phyto-haemagglutinin (PHA) abrogates the restriction observed in the CD127+ Tm cells. We found that infected memory CD4+ T cells from both unstimulated and stimulated HLACs lacked CD127 expression (S2 Fig). However, the stimulation procedure itself downregulated CD127 expression in the memory CD4+ T cells in the uninfected culture; this was also observed in stimulated PBMCs (S2 Fig). Since stimulation-induced CD127 downregulation makes it difficult to assess whether a post-entry restriction mechanism is in play that can be relieved upon T cell activation, we focused directly on assessing SAMHD1 restriction factor function in the CD127+ Tm cells.

## Post-entry restriction of HIV by SAMHD1 is not at play in CD127+ Tm cells

Resting CD4+ T cells from blood restrict HIV infection largely through SAMHD1 [3, 4], a phosphohydrolase whose activity has been shown to be negatively regulated through phosphorylation by IL7 signaling [15]. Because SAMHD1 is preferentially expressed in regions of the lymph nodes where CD127+ Tm cells are prominent [3, 10, 13], we investigated whether it may play a role in limiting HIV infection of tonsillar CD127+ Tm cells. To assess expression levels of SAMHD1 in different subsets, we used a commercial antibody (S3A Fig) to monitor by FACS the expression levels of this protein in CD127+, CD57+, and CD57-CD127- Tm cells from HLACs. Expression levels were similar between the three subsets (S3B Fig), demonstrating that SAMHD1 levels do not correlate with the permissivity of the different memory CD4 + T subsets to HIV infection. Similar conclusions were drawn when SAMHD1 levels were monitored by Western blot, which revealed SAMHD1 levels to be low in the Tm subsets relative to Tn cells (S3C Fig). SAMHD1 is negatively regulated via phosphorylation at Thr592 by Cyclin A2/CDK1, and total protein levels may not fully reflect its ability to restrict HIV infection. We therefore also examined SAMHD1 phosphorylation at Thr592. While THP1 cells harbored high amounts of phosphorylated SAMHD1 as expected [16], all three Tm subsets exhibited low levels of this post-translational modification (Fig 2A). Therefore, neither SAMHD1 nor phospho-SAMHD1 abundance correlates with the permissiveness of CD127+, CD57+, and CD57-CD127- Tm cells to HIV infection.

As a complementary approach to determine whether SAMHD1 plays a role in infection restriction in the CD127+ Tm cells, we abrogated its activity with Vpx, an accessory protein from SIV and HIV-2 that degrades SAMHD1 [17, 18]. To introduce Vpx into F4.HSA HIV-1 virions, we co-transfected 293T producer cells with an F4.HSA proviral construct and a plasmid encoding a Vpx-Vpr fusion construct [19] at two different ratios. Vpx-containing virions were then recovered from the 293T supernatants. To confirm the activity of these viral batches, we compared SAMHD1 levels in CD4+ T cells from unstimulated PBMCs following incubation with no virus, with F4.HSA, and with the two batches of F4.HSA containing Vpx. Both batches of virus containing Vpx diminished the levels of SAMHD1 (Fig 2B). When these same

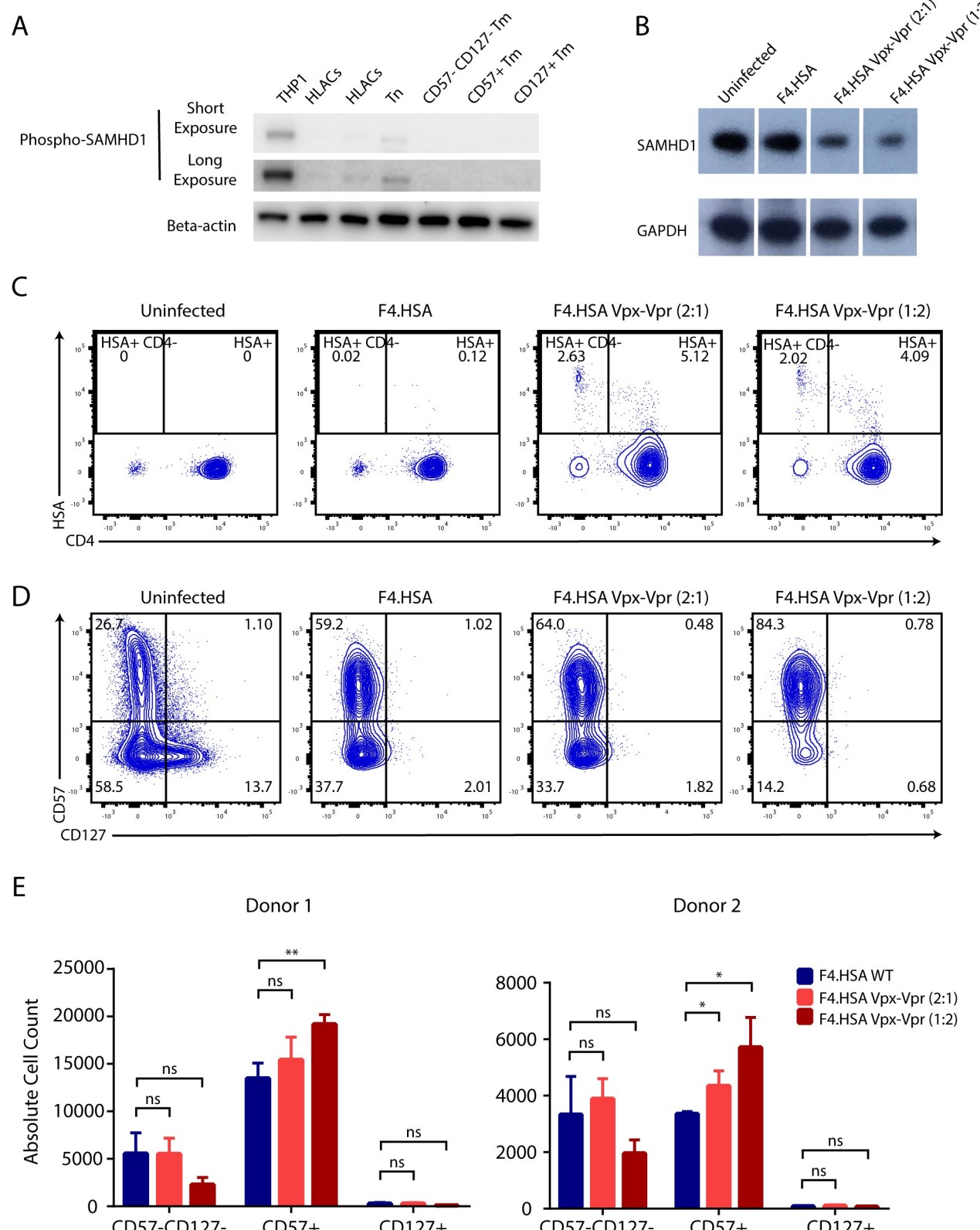

**Fig 2. Relieving SAMHD1 restriction does not increase permissiveness of CD127+ Tm cells to productive infection by HIV-1.** A) The HLAC Tm subsets express low levels of phospho-SAMHD1. Amounts of SAMHD1 phosphorylated at Thr592 in THP1 cells, total HLACs, Tn cells, and the three Tm subsets, as determined by Western blot. Beta-actin served as a loading control. Both short and long exposures of the phospho-SAMHD1 blots are

shown. B) Vpx degrades SAMHD1. Amounts of total SAMHD1 in unstimulated PBMCs that were left uninfected, or infected for 3 days with F4.HSA or F4.HSA harboring Vpx-Vpr fusion protein, as determined by Western blot. 2:1 and 1:2 refer to the ratios of F4.HSA provirus to Vpx-Vpr plasmid in the co-transfections. GAPDH served as a loading control. C) Vpx increases permissiveness of resting PBMCs to productive infection by F4.HSA. Unstimulated PBMCs were mock-treated or exposed for 3 days to F4.HSA lacking Vpx, or harboring different amounts of Vpx. Infection levels were then assessed by flow cytometry. Results are pre-gated on live, singlet CD3+CD8-CD45RO+CD45RA- cells. D) Vpx does not increase HIV infection of CD127+ Tm cells. Representative flow cytometric plots of HLACs mock-treated or exposed for 3 days to F4.HSA lacking or harboring Vpx as indicated. Expression levels of CD57 and CD127 were assessed in uninfected memory CD4+ T cells (plot on left), and HIV-infected (HSA+) memory CD4+ T cells (3 plots on right). Results are pre-gated on live, singlet CD3+CD8-CD45RO+CD45RA- cells. E) Vpx does not increase HIV infection of CD127+ Tm cells. Bar graphs comparing the absolute cell counts of CD57+, CD127+, and CD57-CD127- Tm cells amongst HIV-infected (HSA+) memory CD4+ T cells in two independent donors. Absolute cell counts were determined by normalizing the flow cytometric data to AccuCount beads run for each sample. Error bars correspond to experimental triplicates for each donor. *p<0.05, **p<0.01 as determined using a 2-tailed unpaired student's t-test.

cells were assessed for their ability to support productive infection, cultures treated with virions harboring Vpx were infected at higher rates than those lacking Vpx (Fig 2C). Quantitation of the infection data demonstrated that the presence of Vpx resulted in up to 55-fold higher overall infection rates as defined by the proportion of total HSA+ cells. Up to 132-fold higher infectivity were observed in the Vpx-treated samples when infected cells were defined based on the proportion of HSA+ cells that had downregulated cell-surface CD4, a hallmark of late-stage infection [20].

Having demonstrated that our batches of Vpx-containing HIV-1 virions were active, we used them to infect HLACs and then assessed the distribution of memory CD4+ T cells in the infected cells. Uninfected memory CD4+ T cells exhibited the expected distribution of CD57+, CD127+, and CD57-CD127- Tm cells, while F4.HSA-infected cells were almost exclusively of the CD57+ Tm and CD57-CD127- Tm phenotypes. Although Vpx increased the proportion of CD57+ Tm cells infected by F4.HSA, it did not increase the proportion of infected CD127+ Tm cells (Fig 2D and 2E). These results demonstrate that Vpx does not relieve the viral restriction present in CD127+ Tm cells. In contrast, Vpx activity modestly enhances infection in CD57+ Tm cells, likely reflecting degradation of the SAMHD1 restriction factor. To confirm that Vpx increased infection of CD57+ Tm and did not simply upregulate expression of the CD57 antigen, we sorted the three populations of Tm cells and exposed them to the virions containing Vpx. FACS analysis revealed that Vpx did not increase expression of CD57 in any of the sorted populations (S4 Fig), thereby confirming the increased permissivity of CD57+ Tm in the presence of Vpx.

## CD127+ Tm cells preferentially support latent HIV-1 infection

Having demonstrated that SAMHD1-mediated restriction is likely not the mechanism restricting productive infection in tonsillar CD127+ Tm cells, we tested the possibility that the block occurs later in the viral life cycle, after integration and provirus formation. HLACs were exposed to F4.HSA for 3 days, and then sorted to isolate the CD127+, CD57+, and CD57-CD127- Tm populations. The sorted cells were then subjected to a two-step Alu-Gag ddPCR method similar to those described [21, 22] to quantitate the levels of integrated HIV DNA (Fig 3A and 3B). Data were normalized to mitochondrial content, which was similar between the three sorted populations of cells (S5 Fig). This method has previously been implemented to normalize for levels of integrated HIV-1 provirus [21, 23]. As expected, the sorted CD127+ Tm cells supported low levels of productive infection relative to the sorted CD57+ Tm and CD57-CD127- Tm cells (Fig 3C). However, they harbored the highest levels of integrated HIV DNA (Fig 3D). Repeating this experiment with cells from a total of 5 independent donors revealed that the levels of integrated HIV DNA in the CD127+ Tm cells varied from donor to donor, and were sometimes higher than that in CD57+ and CD57-CD127- Tm cells

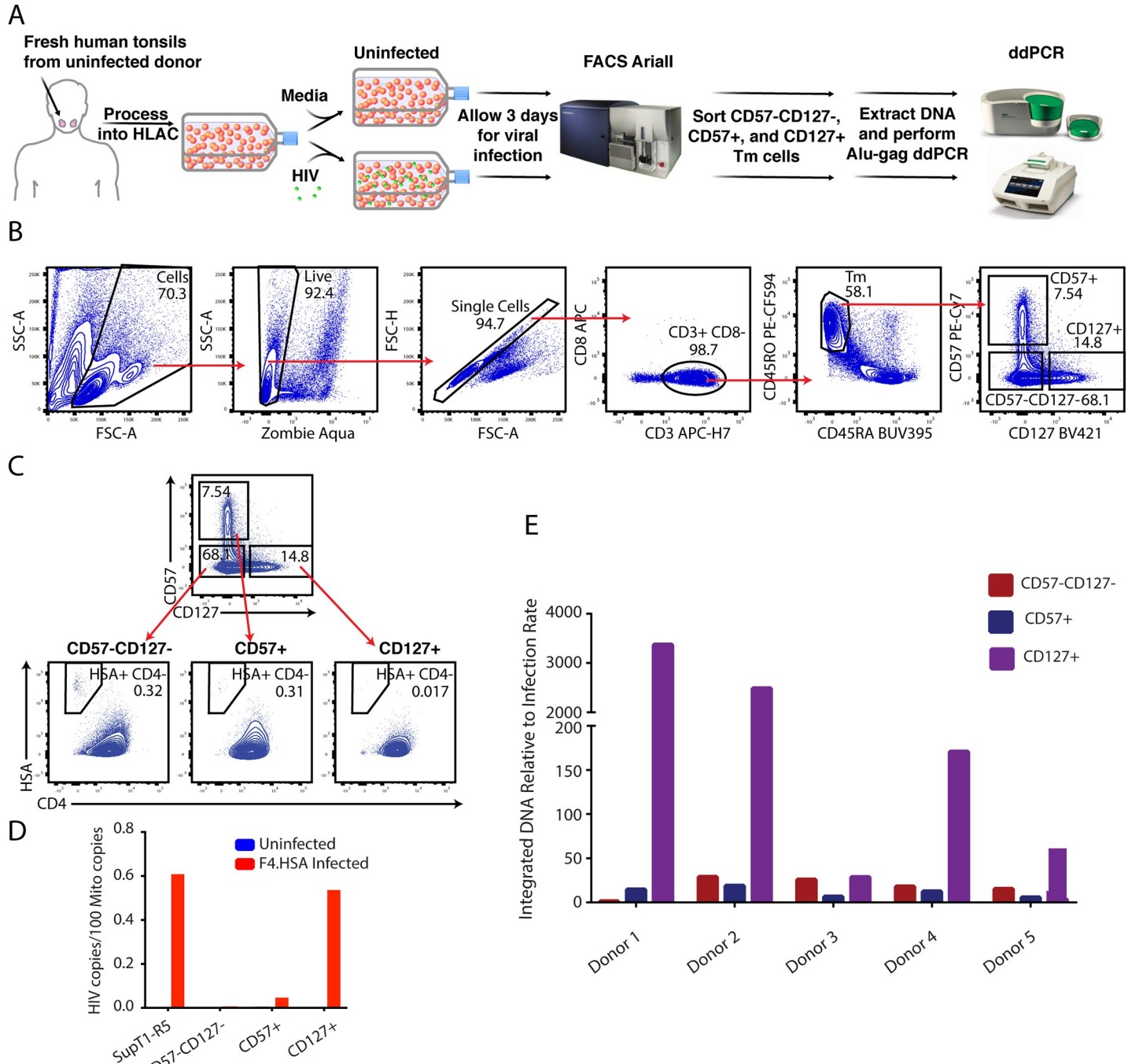

**Fig 3. CD127+ Tm cells preferentially support latent infection by HIV-1.** A) Schematic of experimental design for quantitating integrated HIV DNA in memory CD4 + T cell subsets from HIV-exposed HLACs. HLACs were mock-treated or infected with F4.HSA and cultured for 3 days. Cells were then sorted using an AriaII instrument for the CD57-CD127-, CD57+, and CD127+ Tm populations. Genomic DNA was extracted from sorted cells, and a two-step Alu-Gag ddPCR was performed to amplify and quantitate HIV DNA from these samples. A second ddPCR reaction designed to detect mitochondrial DNA was performed in parallel for all samples to quantify DNA input, and was used for normalization. B) Gating strategy for sorting of HLAC cultures. Live, singlet CD3+CD8- cells (corresponding to CD4 + T cells) were further gated on memory cells (CD45RO+CD45RA-), and then divided into populations of CD57+, CD127+, and CD57-CD127- Tm cells as shown. These sorted populations were used to quantitate the levels of integrated HIV DNA. C) Flow cytometric plots showing the sorted populations of memory CD4+ T cells from F4.HSA-exposed HLACs, demonstrating the expected low infection rates in the CD127+ Tm cells as compared to the other two Tm subsets. D) The samples shown in *panel C* were subjected to ddPCR to quantitate the levels of integrated HIV DNA. Infected SupT1-R5 served as a control. Results were normalized to the amount of mitochondrial DNA in each sample. No integrated HIV DNA was detected in uninfected cells subjected to the same protocol. E) The protocol schematized in *panel A* was conducted on 5 independent donors. The levels of integrated HIV DNA in each population (normalized to mitochondrial content) were divided by the rate of productive infection (as defined by the frequency of HSA+CD4- cells) to demonstrate that the CD127+ Tm cells always harbored a disproportionately high level of HIV DNA relative to their productive infection rate.

and sometimes lower. Importantly, however, the ratios of integrated HIV DNA to productive infection rates as defined by HSA expression were always disproportionately higher in the CD127+ Tm cells than in the other two subsets of memory CD4+ T cells (Fig 3E). Although the Alu-Gag ddPCR assay cannot distinguish intact from defective provirus, given that our *in vitro* culture system is relatively short-term and not subject to immune-mediated pressures, it is likely that most of the sequences we are detecting are intact. These results suggest that the mechanism by which CD127+ Tm cells restrict productive infection by HIV occurs post-integration, and that CD127+ Tm cells preferentially support a latent infection.

To determine to what extent CD127+ Tm cells in tissues harbor latent cells *in vivo*, we obtained sigmoid biopsies from an ART-suppressed, HIV-infected individual as a source of tissue CD127+ Tm cells. CD127+, CD57+, and CD57-CD127- Tm cells were sorted by flow cytometry (S6A Fig). Because the numbers of CD57+ Tm and CD57-CD127- Tm cells were low, these two populations were combined into a single CD127- Tm population. The CD127 + Tm and CD127- Tm cells were assayed for total HIV DNA levels. Both CD127- Tm and CD127+ Tm cells were a source of HIV DNA, and both subsets harbored $> 10^4$ HIV DNA copies per million cells, similar to levels previously reported in gut [24]. Although these results do not suggest preferential persistence of HIV in tissue-derived CD127+ Tm cells, they nonetheless demonstrate that these cells can serve as a cellular source of HIV persistence *in vivo*.

## CD127+ Tm cells exhibit a transcriptional profile of T cell quiescence

To characterize the features of CD127+ Tm cells that bias them towards HIV latency, we conducted RNA-seq profiling of uninfected CD127+, CD57+, and CD57-CD127- Tm cells isolated from three donors. Principal component analysis (PCA) revealed that CD127+ Tm cells were phenotypically distinct from CD57+ Tm cells and CD57-CD127- Tm cells. While the third principal component (PC3) distinguished samples based on donor-dependent differences, principal components 1, 2, and 4 (PC1, PC2, and PC4) together grouped the CD127 + Tm cells into a unique region of the principal component space (Fig 4A).

Thousands of genes were differentially expressed between the CD127+ Tm cells and CD57 + Tm cells or CD57-CD127- Tm cells, including, as expected, the gene encoding CD127 (Fig 4B, S1 Table and S2 Table). Among the top genes preferentially expressed in the CD57+ Tm cells or CD57-CD127- Tm cells relative to the CD127+ Tm cells were TIGIT and CD25, two markers of T cell activation preferentially expressed on HIV-infected cells [25, 26] (Fig 4B). By manually curating the data, we identified a collection of genes known to be associated with HIV-permissive cells that were preferentially downregulated in the CD127+ Tm cells as compared to either of the other subsets (Fig 4C). These included canonical genes associated with Th1 cells (T-bet, IFNγ), a subset highly permissive to HIV infection and stably maintained during antiretroviral therapy [27]; Ox40 and BCL2, which can promote survival of HIV-infected cells [10, 28, 29]; cyclinT1, necessary for efficient elongation of HIV transcripts [30]; metabolic genes (e.g., MTOR) that promote HIV transcription [31]; and CD30 expressed on cells with transcriptionally active HIV [32]. Most of these genes were more highly expressed in either CD57+ Tm cells or CD57-CD127- Tm cells (but not both) than the CD127+ Tm cells. We also identified 569 genes shared between CD57+ Tm cells and CD57-CD127-Tm cells that were differentially expressed relative to CD127+ Tm cells (Fig 4D). The full list of genes differentially expressed between CD127+ Tm cells and CD57+ Tm cells or CD57-CD127- Tm cells are presented in S1 Table and S2 Table.

For a more global and unbiased analysis of the RNAseq data, we conducted ingenuity pathway analysis (IPA). This analysis confirmed that relative to the other two subsets, CD127+ Tm cells indeed down-regulated gene pathways associated with active transcription of HIV (Fig

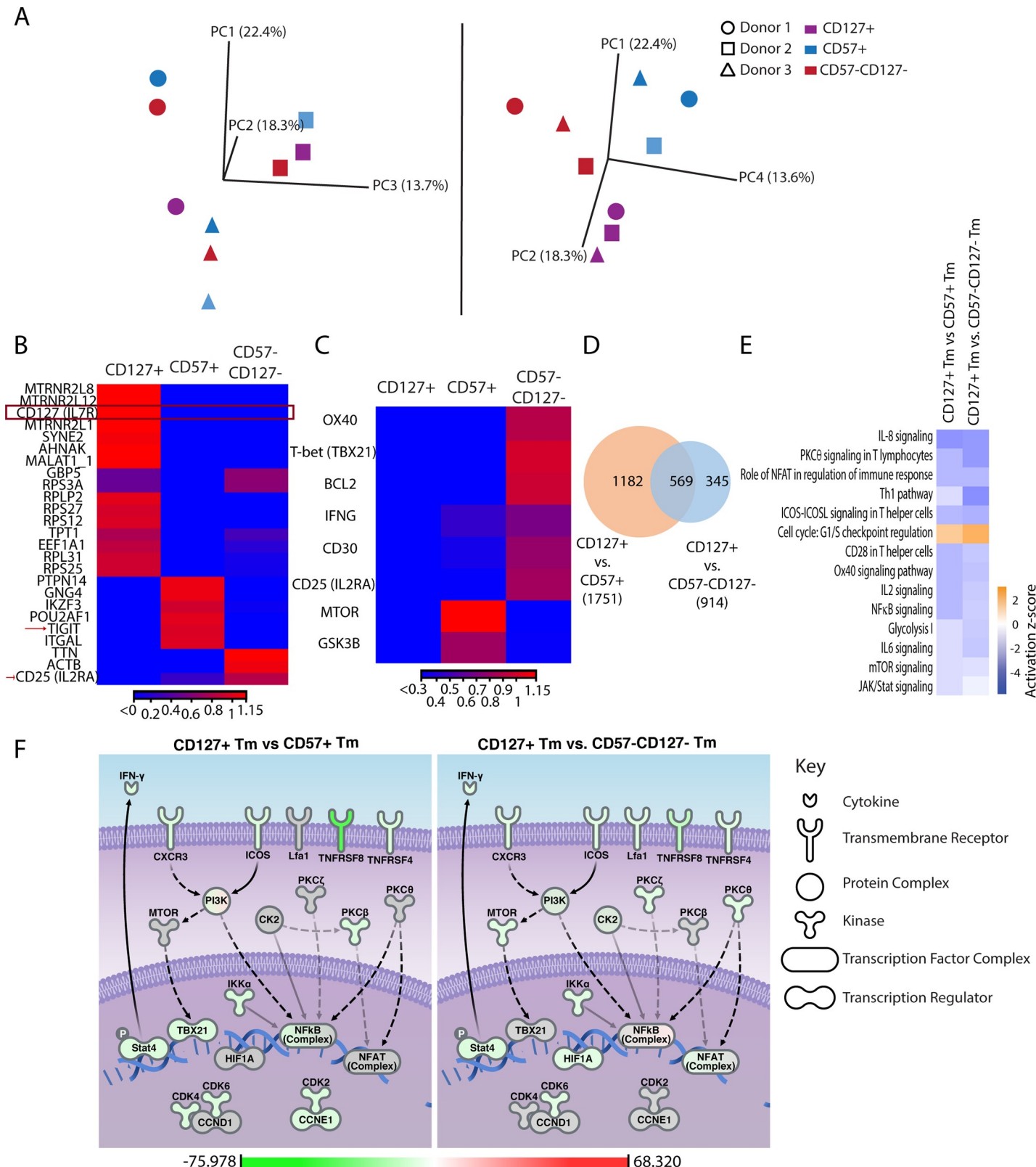

**Fig 4. CD127+ Tm cells exhibit a transcriptional profile of quiescence distinct from that of other tonsillar memory CD4+ T cells.** A) Principal Component Analysis (PCA) of RNAseq results from CD127+ (*purple*), CD57+ (*blue*), and CD57-CD127- (*red*) Tm cells displaying the first 3 principal components (*left*) or the first,

second, and fourth components (*right*). PC3 segregates the samples by donor (each represented by a different shape), while PC1, PC2, and PC4 segregate them by cell type. B) Heatmap of z-scores illustrating the 25 genes with the highest coefficients of variation when comparing CD127+, CD57+, and CD57-CD127- Tm cells. The CD127 transcript is boxed to highlight preferential expression of this gene in the CD127+ Tm cells as expected. Arrows highlight TIGIT and IL2RA, antigens known to be expressed on HIV-infected cells. C) Heatmap of Z-scores illustrating select genes differentially expressed between CD127+, CD57+, and CD57-CD127- Tm cells. These genes were chosen based on known positive associations with HIV infection, and were all expressed at lower levels in CD127+ Tm cells relative to CD57+ Tm cells or CD57-CD127- Tm cells. D) Venn diagram showing the number of overlapping genes differentially expressed between CD127+ and CD57+ Tm cells, and between CD127+ and CD57-CD127- Tm cells. E) Activation z-scores of select IPA pathways as compared between CD127+ and CD57+ Tm cells, or CD127+ and CD57-CD127- Tm cells. Pathways are listed according to z-score, from top to bottom (highest to lowest absolute z-score) with a p-value cutoff filter of $\log_{10}1.3$. Blue corresponds to pathways downregulated in CD127+ Tm cells, with the intensity of the blue corresponding to the extent of downregulation. Orange corresponds to pathways upregulated in CD127+ Tm cells. For a full list of significant pathways, see S3 Table. F) Graphical representation of select pathways differentially active in CD127+ Tm cells as compared to CD57+ (*left*) or CD57-CD127- (*right*) Tm cells. Biological relationships between the different pathway components were found by IPA and are represented as dashed lines if the relationship is indirect and continuous lines if the relationship is direct. Pathway components are displayed using various shapes that represent the functional class of the gene product, as defined in the figure key. The names of the pathway components correspond to those in the IPA Knowledge Database. Pathways components were colored according to experimental expression values, with the fold-change color code displayed at the bottom of each panel. These data demonstrate that multiple pathways associated with T cell activation and HIV gene expression are downregulated in CD127+ Tm cells.

4E). In particular, NF-κB and NFAT signaling, known to play important roles in HIV gene transcription [33–35], were decreased. Other examples of cellular pathways associated with HIV transcriptional activity that were dampened in the CD127+ Tm cells are Ox40 signaling, glycolysis, and JAK/Stat signaling. A more in-depth view of the genes involved in the major suppressed pathways is shown in Fig 4F, which highlights the following: 1) There was suppression of activators of NF-κB activity (CK2, PKCζ, PKCθ) and NFAT activity (PKCθ) in CD127+ Tm cells relative to CD57+ Tm cells, and suppression of an activator of NFAT activity (PKCβ) in CD127+ Tm cells relative to CD57-CD127- Tm cells. Dampened PKCθ, PKCζ, PKCβ, together with low levels of genes associated with Th1 signaling (Stat4, IFN-γ, TBX21 (Tbet), and CXCR3), also suggest a state of low T cell activation in the CD127+ Tm cells relative to the other subsets; 2) There was an overall suppression of active cellular metabolism, as demonstrated by low PI3K or mTOR levels, in the CD127+ Tm cells relative to the other subsets; 3) There was an overall lack of active cellular proliferation in the CD127+ Tm cells (consistent with upregulation of G1/S checkpoint regulation, Fig 4E), as indicated by lower expression levels of CK2, CDK2, CDK4, CDK6 and CCNE1 (cyclin E), which are all positive regulators of cell cycle progression [36, 37]. Also depicted in the figure are the low expression levels of HIF1α, a known activator of the HIV LTR [38], and TNFR family members (CD30, Ox40) associated with transcriptionally active HIV [10, 28, 32], in the CD127+ Tm cells. In summary, the RNAseq and IPA analyses suggest that compared to other memory CD4+ T cell subsets, the CD127+ Tm cells reside in a quiescent state that may limit HIV transcription.

## Integration site analysis in latently-infected CD127+ Tm cells

The RNAseq data suggest that the provirus in latently-infected CD127+ Tm cells may be transcriptionally silenced through a state of cellular quiescence. Such a state of quiescence may lead to HIV integrating into closed or inaccessible regions of the genome. To test this possibility, we determined the sites of F4.HSA integration in latently-infected CD127+ Tm cells. HLACs from two donors were exposed to F4.HSA for 3 days, and then sorted for latent (HSA-) CD127+ Tm cells for integration site analysis (Fig 5A). Latently-infected CD57+ Tm cells were sorted for comparison. Of the 1,207 unique proviral integration sites that were identified, the vast majority were in genic regions (S7A Fig), which tend to reside within open regions of the genome [39, 40]. Comparison of integration sites in the CD127+ vs. CD57+ Tm cells did not reveal any significant differences in the frequencies of integration into genic vs. non-genic regions. To more directly assess whether HIV preferentially integrated into inaccessible regions of the genome in CD127+ Tm cells, we mapped our integration sites against publicly-available datasets of chromatin accessibility. Mapping the sites against a total of 127

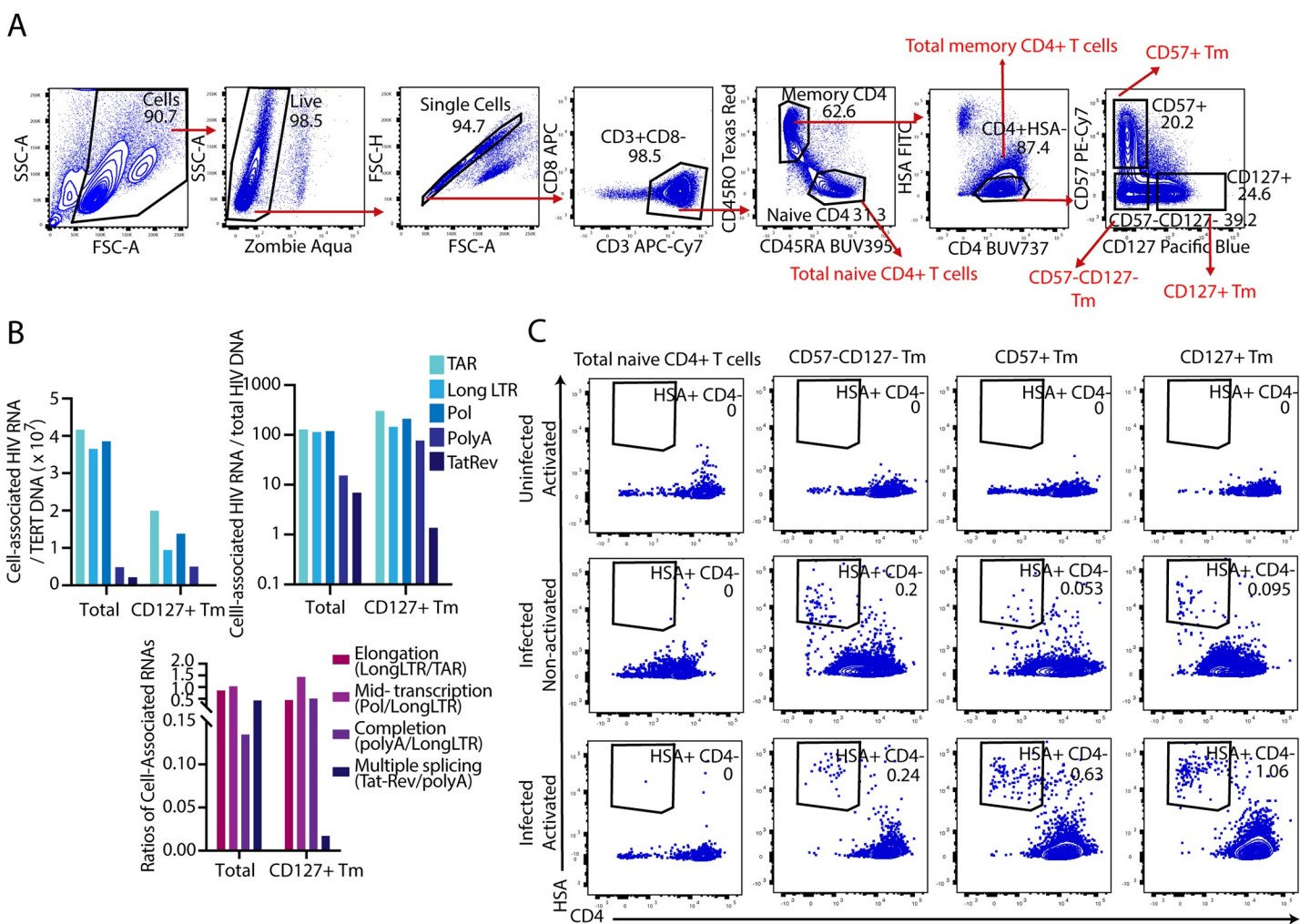

**Fig 5. Latently-infected CD127+ Tm cells can be reactivated by T cell stimulation.** A) Gating strategy for sorting of HLAC cultures. Live, singlet CD3+CD8- cells (corresponding to CD4+ T cells) were further gated on memory (CD45RO+CD45RA-) or naïve (CD45RO-CD45RA+) cells. The latently-infected CD127+, CD57+, and CD57-CD127- Tm cells were then isolated by gating on the CD4+HSA- cells as shown. B) Latently-infected CD127+ Tm cells can transcribe HIV but are inhibited in HIV splicing. Total (TAR), 5' elongated (R-U5/pre-Gag "Long LTR"), Pol, polyadenylated (PolyA), and multiply-spliced Tat-Rev (TatRev) HIV RNAs were measured in the total infected culture, or in the HSA- CD127+ Tm cells sorted as described in *panel A*. Data are normalized to the housekeeping gene TERT (*top left*) or to the levels of HIV DNA in each sorted population (*top right*). *Bottom*: The extent of elongation, mid-transcriptional elongation, transcript completion, and splicing were determined by examining the ratios of the indicated transcripts. The ratio of Tat-Rev/polyA transcripts in latently-infected CD127+ Tm cells was disproportionately low, suggesting a defect in HIV splicing in these cells. Shown are results of one of two representative donors. C) The sorted populations of naïve CD4+ T cells, as well as CD57-CD127-, CD57+, and CD127+ Tm cells defined in *panel A* from uninfected or infected HLAC cultures were mock-treated or stimulated with anti-CD3/CD28 beads and then assessed levels of reactivation three days later. The proportions of infected (HSA+) cells that have downregulated cell-surface CD4 are indicated. The infected cells in the non-activated samples are expressing HSA due to spontaneous reactivation of the sorted HSA- cells, while those in the activated samples correspond to stimulation-induced reactivation. Shown are results of one of four representative donors, with the average induction in infection rates, in the CD127+ Tm cells, between the non-activated and activated cultures being 6.8-fold (range 4.4–11.2-fold).

DNase-seq and ATAC-seq datasets generated from primary human CD4+ T cells (see Methods) revealed a trend towards more frequent integration into closed, inaccessible regions in the CD127+ Tm subset than the CD57+ Tm subset; this trend however was not statistically significant (S7B Fig).

## Proviruses in CD127+ Tm cells are reactivated following T cell stimulation

To further characterize to what extent the CD127+ Tm were quiescent, we determined to what extent these cells have silenced HIV transcription. HLACs were again exposed to F4.HSA for 3 days, and then sorted for latently-infected (HSA-) CD127+ Tm cells (Fig 5A). A panel of RT-ddPCR assays was implemented to quantify various HIV transcripts as previously described [6]. Surprisingly, initiated, elongated, completed, and spliced transcripts were all detected in the latent CD127+ Tm cells, although at lower levels than observed in the total population of infected cells (Fig 5B). Comparing the ratios of different HIV transcripts revealed that HIV transcription in the latently-infected CD127+ Tm cells was blocked at the step of viral RNA splicing, since the ratio of multiply-spliced to completed transcripts was markedly lower in these cells than that observed in the total population of cells from the infected culture (Fig 5B).

These results suggest that latently-infected CD127+ Tm cells include cells that that are transcribing HIV, and that these cells therefore are unlikely to exist in a state of latency refractory to reactivation [41]. To test this directly, HLACs were mock-treated or exposed to F4.HSA for 3 days and then sorted for HSA- CD127+, CD57+, or CD57-CD127- Tm cells (Fig 5A). As a negative control, naïve CD4+ T cells, which are poorly susceptible to infection by F4.HSA due to their low fusogenicity with CCR5-tropic HIV-1 [10], were also sorted. The same subsets of cells sorted from uninfected HLACs served as additional negative controls. All sorted populations of cells were either mock-treated or stimulated with anti-CD3/CD28 beads, and monitored 3 days later by flow cytometry for the levels of productively-infected cells. As expected, no infected cells were observed in sorted populations from uninfected cultures. While naïve CD4+ T cells exhibited minimal reactivation, the sorted HSA- CD127+ Tm cells exhibited a clear population of reactivated cells upon stimulation (Fig 5C). The sorted HSA- CD57+ Tm cells were also efficiently reactivated upon stimulation, while the unstimulated and stimulated HSA- CD57-CD127- Tm cells exhibited similar levels of reactivation, suggesting high levels of spontaneous viral reactivation in this population. Other latency reversal agents, in particular the ingenol PEP005, could also reverse latency in CD127+ Tm cells, while the HDAC inhibitor romidepsin did not (S8A Fig). IL7 itself, the ligand for CD127, was capable of reactivating HIV from latently-infected CD127+ Tm cells in some but not all donors (S8B Fig). In conclusion, these data demonstrate that latent HIV proviruses within CD127+ Tm cells are capable of reactivating upon stimulation.

## Discussion

In this study we identify a population of tissue-specific memory CD4+ T cells expressing the IL-7 receptor-alpha chain that upon exposure to HIV preferentially supports latent infection by the virus. These findings follow up on our prior study where we described efficient fusion of CCR5-tropic F4.HSA HIV-1 to tonsillar CD127+ Tm cells but little or no productive infection [10]. In that study, we conducted two sets of analyses demonstrating that the lack of CD127 expression on HIV-infected cells was not due to downregulation of the receptor by the virus. First, a bioinformatics approach [42] based on 38-parameter CyTOF phenotyping datasets was used to show that HIV-infected cells did not harbor any features of CD127+ Tm cells, but instead harbored features of CD57+ Tm cells and CD57-CD127- Tm cells. Second, experimental data demonstrated that purified CD127+ Tm cells were efficient at fusing to HIV but poorly supported productive infection [10]. The purpose of the current study was to determine the molecular basis underlying the inability of CD127+ Tm cells to support productive infection.

We first considered the possibility that viral restriction by SAMHD1 was preventing the early steps of the viral life cycle in CD127+ Tm cells. One rationale for testing this idea came

from a recent study demonstrating that SAMHD1 restriction in memory CD4+ T cells from blood could be relieved by treatment with IL7, the ligand for CD127 [15]. Although expression of CD127 was not assessed in that study, it seemed likely that CD127-expressing cells were the ones responding to IL7. We therefore considered the possibility that SAMHD1 may also restrict infection in tissue-derived CD4+ T cells. In line with the notion that SAMHD1 may be present in CD127+ Tm cells are prior observations that SAMHD1 is detected in the extrafollicular regions of the lymph node where these cells reside [3, 10, 13]. Our findings, however, suggest that SAMHD1 restriction is unlikely the mechanism preventing productive infection of CD127+ Tm cells, since active SAMHD1 was expressed at similar levels in CD127+ Tm cells and the highly susceptible CD57+ and CD57-CD127- Tm subsets. More importantly, Vpx-mediated degradation of SAMHD1 did not rescue HIV infection of CD127+ Tm cells, but instead increased the proportion of CD57+ Tm cells that were productively infected. This suggests that restriction by SAMHD1 is not occurring in CD127+ Tm cells, but rather in other memory CD4+ T cell subsets. Further supporting the notion that early post-entry restriction of HIV by SAMHD1 is not the mechanism in play were our findings that HIV-exposed CD127 + Tm cells in fact harbored relatively high levels of integrated DNA. The fact that CD127+ Tm cells could harbor higher levels of HIV DNA than CD127- Tm cells while expressing low levels of the LTR-driven reporter gene suggest a propensity of this subset to undergo latent infection by HIV.

We had previously characterized by CyTOF the phenotypic differences between the nonpermissive CD127+ Tm cells and the permissive CD57+ Tm cells, and found that these subsets were phenotypically distinct in a manner independent of CD127 and CD57 expression; in other words, these two subsets clustered distinctly even when CD57 and CD127 were not used as analysis parameters. To use a more unbiased approach to characterize the CD127+ Tm cells, we implemented global gene expression profiling of CD127+ Tm cells relative to both CD57+ Tm cells and CD57-CD127- Tm cells. Consistent with the phenotypic uniqueness of CD127+ Tm cells were our findings that these cells resided in their own region of principal component space. Relative to the other two subsets, these cells expressed transcripts associated with cellular quiescence. Pathway analysis revealed that both NF-κB and NFAT signaling were diminished in the CD127+ Tm cells in a manner associated with downregulation of PKC family members PKCθ, PKCζ, PKCβ, and/or CK2, known activators of the NF-κB or NFAT signaling [43, 44]. Both NF-κB and NFAT have long been recognized as signaling pathways important for elongation of HIV transcripts [45]. Interestingly, recent studies have implicated glycolytic pathways as important mediators of HIV gene transcription [46, 47] and, more specifically, activation of components of the mTOR pathway [48]. Our results demonstrated that these pathways were also suppressed in the CD127+ Tm cells. Select genes previously associated with transcriptionally active HIV, such as CD30, CD25, and Ox40 [10, 26, 28, 32], were also suppressed amongst the CD127+ Tm cells. These results suggest that CD127+ Tm cells may support latent infection because they lack expression of multiple factors that allow for efficient HIV gene transcription. They may also preferentially express factors that actively promote HIV latency. For example, *c-Myc* was upregulated 7.7-fold in CD127+ Tm cells relative to CD57+ Tm cells, and 2.2-fold relative to CD57-CD127- Tm cells. A prior study implicated c-Myc in promoting viral latency by recruiting HDAC1 to the HIV-1 LTR to inhibit Tat-activated HIV-1 LTR expression [49]. Furthermore, latency reversal of patient-derived cells with the HDAC inhibitor valproic acid was accompanied by c-Myc downregulation. To what extent HIV latency in CD127+ Tm cells is mediated by c-Myc requires follow-up studies.

Interestingly, the overall landscape of cellular quiescence in CD127+ Tm cells relative to CD57+ Tm cells was associated with a trend toward higher frequency of HIV integrating into inaccessible regions of the genome as defined by DNAse-seq and ATAC-seq data. Of note

however, DNAse-seq and ATAC-seq derived chromosome accessibility data are biased towards promoter regions, particularly those of highly expressed genes [50], and promoter regions in general are not known to be preferential targets of HIV integration. Consistent with this, the proviral integration sites we identified were mostly within inaccessible genomic regions in both the CD57+ and CD127+ Tm cells. At the same time, we found that HIV preferentially integrated into genic locations in both subsets, with no preferential genic integration in one subset over the other. Taken together, these data suggest that in our *in vitro* system, latently-infected tissue cells primarily harbor provirus in non-promoter regions of genes. Whether the overall landscape of cellular quiescence in the CD127+ Tm cells causally drives this integration pattern of HIV requires further investigation. Despite this state of cellular quiescence, however, our transcriptional profiling of latently-infected CD127+ Tm cells revealed that some HIV transcription was occurring, but that splicing of these transcripts was blocked. Future studies are warranted to better characterize the contribution of splicing defects in maintaining HIV latency in these cells.

Although to our knowledge CD127 has not previously been shown to be preferentially expressed on latent cells, its ligand IL7 has been implicated in both latency establishment and maintenance. A recent study demonstrated that upon treatment of blood-derived resting CD4 + T cells with IL7, SAMHD1 was rapidly phosphorylated at Thr592 and thereby inactivated, permitting increased reverse transcription and integration of HIV [15]. A role for IL7 in latent cell maintenance *in vivo* is supported by the observation that higher IL7 levels in the blood of HIV-infected, ART-suppressed individuals associate with a slower decrease in reservoir size over time [14]. However, both of these studies only analyzed cells from blood. Our findings suggest that different mechanisms may be at play in tissue. We propose that unlike in blood, lymphoid CD127 + Tm cells do not exhibit early post-entry restriction by SAMHD1, but rather readily allow for integration but not HIV gene expression. While IL7 appears to promote HIV latency establishment in resting cells derived from blood, the tissue CD127+ Tm cells are poised to undergo latent infection in the absence of IL7, as IL7 was not added to our cultures during latency establishment and is only present in limited amounts in HLACs [51]. Importantly, the IL7-responding latent cells in blood [15] as well as the lymphoid CD127+ Tm cells studied here are both capable of reactivating upon stimulation, suggesting that both populations can contribute to viral rebound during antiretroviral treatment interruption. Furthermore, IL7 itself can promote reactivation of the latent lymphoid CD127+ Tm cells in some albeit not all donors; it is possible that a more consistent effect across donors was not observed because IL7 only weakly stimulates LTR-driven gene expression [52–54]. It is likely that the latent CD127+ Tm cells in tissues, like those in blood, are also maintained by IL7-driven homeostatic proliferation. Future studies characterizing the role of IL7 in maintaining this latent reservoir are warranted, and will require use of pre-sorted populations of CD127+ Tm cells, since CD127 is downregulated upon IL7 treatment [55].

The CD127+ Tm cells are distinct from the PD1+ and PD1-CTLA4+ memory CD4+ T cell populations that have been shown to harbor latently-infected cells in tissues *in vivo* [56, 57]. The latently-infected PD1+ memory CD4+ T cells are primarily T follicular cells (Tfh) [56], which overlap with the CD57+ Tm cells subset in tonsils [10, 58]. These cells are highly susceptible to productive infection [10] but at the same time can also support latent infection. Indeed, we found that stimulation of sorted HSA- CD57+ Tm cells from F4.HSA-exposed HLACs resulted in HIV reactivation (Fig 5C). Although both CD57+ and CD127+ Tm cells can support latent infection, the CD127+ Tm cells are unique in that they *preferentially* undergo latent infection over productive infection. Latently-infected PD1-CTLA4+ memory CD4+ T cells include mostly regulatory T cells (Tregs) and are distinct from the CD127+ Tm cells as they express low levels of CD127 [57]. However, whether these cell types are truly distinct lineages,

or whether latently-infected CD127+ Tm cells may eventually downregulate CD127 (e.g., after receiving IL7 signals [55]) and upregulate CTLA4 (e.g., after stimulation) remains to be tested.

In conclusion, our study suggests that cell-intrinsic features of a specific subset of memory CD4+ T cells can bias them to preferentially support latent infection by HIV. The CD127+ Tm cells we describe here may serve as a useful *in vitro* tissue model of HIV latency. Although multiple models of HIV latency exist [59], all utilize blood-derived cells. Benefits of using *in vitro* infection of CD127+ Tm cells as a model for HIV tissue latency are the ease of use due to the relatively minimal *ex vivo* manipulation of cells as compared to other primary cell models of HIV latency that have been established in blood, which require treating cells with select cytokines or other factors followed by prolonged periods of *ex vivo* culture [59]. Establishment of latently-infected CD4+ T cells using CD127+ Tm cells simply entails infecting HLACs with replication-competent HIV, and then isolating memory CD4+ T cells expressing CD127. Isolating cells based on CD127 is preferable to isolating cells based on PD1 expression, as the latter will include a large population of productively-infected cells. It is also preferable over isolation based on CTLA4 expression, since in the absence of stimulation the vast majority of CTLA4 is intracellular and cell permeabilization is required to quantitate its expression levels [57, 60]. We envision that the CD127+ Tm cell latency model can be used to investigate multiple aspects of tissue latency establishment and maintenance, such as whether the sites of integration are defined by specific chromatin modifications, what factors are responsible for promoting and maintaining latency, and what kinds of latency reversal agents are effective or ineffective in tissues. Furthermore, should CD127+ Tm cells be shown to harbor a significant fraction of the inducible reservoir *in vivo*, targeting this population of cells may be an effective approach to achieve HIV remission.

## Materials and methods

### Cells

HEK293T cells were obtained from the ATCC and cultured in D10 (DMEM supplemented with 10% fetal bovine serum (FBS) (Gemini Bio-Products), 100 U/mL penicillin, and 100 μg/ml streptomycin (Thermo Fisher)). SupT1-R5 cells [12] and THP1 cells (ATCC) were cultured in RP10 (RPMI supplemented with 10% FBS, 100 U/mL penicillin, and 100 μg/ml streptomycin). Human tonsils from the Cooperative Human Tissue Network (CHTN) were processed into human lymphoid aggregate cultures (HLACs) and cultured as previously described [10]. Human peripheral blood mononuclear cells (PBMCs) were isolated through Ficoll-Paque density gradient sedimentation by processing leukoreduction system chambers from Blood Centers of the Pacific. All primary cells were cultured in 96-well U-bottomed polystyrene plates ($10^6$ cells/well) in 200 μL media/well. HLACs were cultured in Tonsil Media (RPMI supplemented with 15% FBS, 100 μg/ml gentamicin, 200 μg/ml ampicillin, 1 mM sodium pyruvate, 1% non-essential amino acids (Mediatech), 1% Glutamax (Thermo Fisher), and 1% Fungizone (Invitrogen)), while PBMCs were cultured in RP10. For analysis of *in vivo* specimens, one HIV-1 subtype-B individual continually suppressed on ART for > 8 years (HIV RNA < 40 copies / ml) who initiated therapy during chronic (>1 year) infection was recruited to donate sigmoid biopsies, which were processed into single-cell suspensions using previously described methods [61]. The CD127+, CD57+, and CD57-CD127- Tm cells were sorted as described in the Flow Cytometry section below. Due to limited cell numbers, the sorted CD57+ and CD57-CD127- Tm populations were combined into a single CD127- Tm population.

## Generation of virions

HEK293T cells were cultured in T175 tissue culture flasks until they reach 80% confluency, at which time they were transfected with a transfection mixture consisting of 30 μg of DNA vectors mixed with 90 μg of the transfection agent polyethylenimine-max (PEI-max) (Polysciences) diluted in Opti-MEM (Gibco). Virions were harvested from culture supernatants 48 h post-transfection and concentrated by ultracentrifugation. The proviral construct used to generate the HIV-1 HSA reporter virus (F4.HSA) is pNL-HSA.6ATRi-C.ZM109F.PB4.ecto as was previously described [10]. The construct for generating Vpx-Vpr fusion protein was previously described [19] and received as a gift from Dr. Nicolas Manel. For generating F4.HSA containing the Vpx-Vpr fusion protein, the F4.HSA proviral construct was co-transfected with the Vpx-Vpr construct at a 2:1 ratio (F4.HSA Vpx-Vpr (2:1)) or 1:2 ratio (F4.HSA Vpx-Vpr (1:2)). Viral titers were measured by the Lenti-X p24$^{Gag}$ Rapid Titer Kit (Clontech).

## Infection assays

HLACs, PBMCs, or SupT1-R5 ($10^6$ cells/mL) were mock-treated or infected with F4.HSA at a final concentration of 1500–3000 ng/mL p24$^{Gag}$, and cultured for 3 days at 37˚C. Where appropriate, primary cells were pre-activated with 10 μg/ml PHA (Sigma) or with Dynabeads Human T-Activator CD3/CD28 (Thermo Fisher), using the appropriate media supplemented with 100 U/mL human interleukin-2 (IL-2) (Life Technologies). To assess infection rates, cells were subjected to flow cytometric analysis as detailed below. For experiments where samples were sorted after infection, cells were first enriched by negative selection for CD4+ T cells using the EasySep Human CD4+ T Cell Enrichment Kit (STEMCELL Technologies) to minimize sort time. Where indicated, sorted cells were mock-treated or stimulated for 2–3 days with Dynabeads Human T-Activator CD3/CD28 (Thermo Fisher) at a ratio of 1 bead/cell in the presence of 60U/ml IL-2 (Life Technologies), 12 nM PEP005 (Ingenol 3-angelate, Sigma-Aldrich), 40 nM Romidepsin (Sigma-Aldrich), or 10 ng/ml human IL-7 (R&D System), and then harvested for flow cytometric analysis as detailed below.

## Flow cytometry

Cells from the *in vitro* HLAC specimens or the gut biopsies were washed once in FACS buffer (PBS + 2% FBS + 2 mM EDTA), and resuspended to a final concentration of 50 million cells/mL. Cells were stained for 15 min at room temperature with Zombie Aqua viability dye (BioLegend) in PBS, and then an additional 30 min at 4˚C with a cocktail of fluorescently-labeled antibodies in FACS buffer. Antibodies were purchased from eBioscience (anti-CD57 (TB01) PE-Cy7), BD Biosciences (anti-CD3 (SK7) APC-H7, anti-CD4 (SK3) BUV737, anti-CD45RA (HI100) BUV395, anti-CD45RO (UCHL1) PE-CF594, anti-CD127 (HIL-7R-M21) BV421, and anti-HSA (M1/69) FITC), and BioLegend (anti-CD3 (SK7) APC-Cy7, anti-CD4 (A161A1) PE-Cy7, and anti-CD8 (SK1) APC). For flow cytometric detection of SAMHD1, anti-SAMHD1 antibody (Bethyl Laboratories) was conjugated to DyLight 594 using the Lightning-Link Rapid DyLight 594 Antibody Labeling Kit (Novus Biologicals). Stained samples were run on a FACS AriaII flow cytometer (BD Biosciences), and data were analyzed in FACSDiva (BD Biosciences) and FlowJo (Treestar). To determine the absolute numbers of cells in samples analyzed by FACS, AccuCount beads (Spherotech) were added to stained samples prior to each FACS run per manufacturer's instructions, and absolute cell counts were calculated based on the known number of beads added.

## Western blot

Cells were lysed with Cell Extraction Buffer (Invitrogen) supplemented with protease and phosphatase inhibitors (Roche) per manufacturer's protocol. Samples were run on NuPAGE 4–12% Bis-Tris Protein Gels (Invitrogen), and then transferred to polyvinylidene difluoride (PVDF) membranes compatible with the iBlot 2 Dry Blotting System (Invitrogen). The membranes were then incubated with either anti-SAMHD1 (Bethyl Laboratories), anti-phospho (Thr592)-SAMHD1 (ProSci), anti-GAPDH (Cell Signaling Technology), or anti-beta-actin (Sigma). After a series of washes, the appropriate secondary antibodies (Horseradish peroxidase (HRP)-conjugated anti-rabbit or anti-mouse immunoglobulin G (GE Healthcare)) were added. Membranes were developed using Immobilon Forte Western HRP Substrate (Millipore).

## Quantification of HIV-1 provirus

To quantify integrated F4.HSA DNA, cellular DNA was extracted from sorted populations using the DNeasy Blood & Tissue Kit (Qiagen). The samples were analyzed using a two-step Alu-gag PCR method similar to methods previously described [21, 22] but adapted for use of droplet digital PCR (ddPCR) as the method of DNA quantification. Integrated DNA levels were normalized to mitochondrial content similar to previously described methods [21, 23]. The sequences for the preamplification primers were as follows: genomic Alu forward, 5′-GCC TCC CAA AGT GCT GGG ATT ACA G-3′; and HIV-1 gag reverse, 5′-GCT CTC GCA CCC ATC TCT CTC C-3′ [22]. The sequences of primers and probes used for ddPCR were as follows: late RT forward, 5′-TGTGTGCCCGTCTGTTGTGT-3′; late RT reverse, 5′-GAGTCCT GCGTCGAGAGAGC-3′; late RT probe, 5'-(6-FAM)CAGTGGCGC(ZEN)CCGAACAGGGA (IBFQ)-3'; mitochondrial forward primer, 5′-ACCCACTCCCTCTTAGCCAATATT-3′; mitochondrial reverse primer, 5′-GTAGGGCTAGGCCCACCG-3′; mitochondrial probe, 5'-(6-FAM)CTAGTCTTT(ZEN)GCCGCCTGCGAAGCA(IBFQ)-3' [21, 23]. All primers and probes were purchased from Integrated DNA Technologies. To pre-amplify DNA samples from sorted cells, 15 μL PCR solution was prepared with 1X Qiagen PCR buffer, 3.75 U Taq DNA Polymerase (Qiagen), 2 mM dNTPs (Thermo Scientific), 100 nM Alu forward primer, 600 nM gag reverse primer, and 75 ng of template. Pre-amplification was carried out using the GeneAmp PCR System 9700 thermocycler (Applied Biosystems) implementing the following cycling conditions: 20 cycles each consisting of denaturation at 93˚C for 30 s, annealing at 50˚C for 1 min, and extension at 72˚C for 5 min. Next, ddPCR was applied using a BioRad QX100 ddPCR Reader to quantify the pre-amplified PCR products. Each 25 μL ddPCR mix comprised the ddPCR Probe Supermix (no dUTP) (Bio-Rad), 250 nM primers, 200 nM probe, 2.5 U HaeIII restriction enzyme (New England BioLabs), and 5 μL pre-amplified product. For mtDNA detection, pre-amplified products were diluted 1:100 prior to the addition of reaction mixes. After droplet generation using the BioRad QX100 Droplet Generator, the following cycling conditions were used on the BioRad C1000 Thermocycler: 10 min at 95˚C, 50 cycles each consisting of 30 s denaturation at 94˚C followed by 61.2˚C extension for 60 s, and a final 10 min at 98˚C. Data were analyzed on QuantaSoft Analysis Pro (Bio-Rad). For quantitation of HIV DNA in the patient samples, total DNA from sorted samples was resuspended in 50 μL of QIAGEN buffer EB. HIV DNA (R-U5-pre-Gag region; "LongLTR") and telomere reverse transcriptase (TERT) were measured in duplicate aliquots of DNA using droplet digital PCR, as previously described [6], with the following modification to account for limited cell numbers available per condition: 9 μL/well total DNA was used to measure the DNA cell equivalents (TERT) and 9 μL/well was used to measure HIV DNA (LongLTR). Levels of HIV DNA were expressed as copies/μg DNA or were normalized to $10^6$ cells as defined by TERT copies.

## HIV integration site analysis

To assess the genomic sites that F4.HSA integrated into, cellular DNA was extracted from sorted HSA- CD127+ or CD57+ Tm cells using the DNeasy Blood & Tissue Kit (Qiagen). For each extract, the BIO-RAD ddPCR QX200 system was used to quantify the total number of HIV DNA copies using forward primer 5'-TCTCGACGCAGGACTCG-3', reverse primer 5'-TACTGACGCTCTCGCACC-3', and probe 5'-/56-FAM/CTCTCTCCT/ZEN/TCTAGCCTC/31ABkFQ/-3' under the following thermocycler conditions: 95°C for 10 minutes, 45 cycles of (95°C for 30 seconds, 60°C for 1 minute) and 98°C for 10 minutes [62]. A total of 5 µL of each sample was processed in triplicates for viral integration site amplification. First, each replicate was subjected to multiple displacement amplification (MDA) using the REPLI-g Single Cell Kit (Qiagen) to amplify input template signals. Then, each replicate was subjected to viral integration site amplification using the Lenti-X Integration Site Analysis Kit (Clontech Takara) and according to established protocols [63]. This was followed by MiSeq (Illumina) deep sequencing conducted at the Massachusetts General Hospital DNA Core. The resulting FASTQ files were bioinformatically processed using an in-house pipeline. Briefly, FASTQ files were filtered for reads containing viral-host junctions using NCBI blast+ [64], and then mapped to the human reference genome hg38 using BLAT [65]. Because MDA amplified input template copy numbers, identical integration sites in each sample were collapsed into single observations. To define genic versus non-genic viral integration sites, the integration site coordinates were mapped to the NCBI Reference Sequence (RefSeq) database [66]. To assess whether any of the integration sites occurred in genomic regions previously shown to be accessible, we mapped our integration sites against publicly-available human CD4+ T cell datasets of chromatin accessibility. We combined 13 samples of DNase-seq data (ENCODE Hotspot accession numbers 'ENCFF758ERD','ENCFF927XKI','ENCFF301QTV','ENCFF313IZW','ENCFF446RUX','ENCFF122YQT','ENCFF189ETX','ENCFF746XGC','ENCFF856JUE','ENCFF066INP','ENCFF202YKQ','ENCFF851WFX','ENCFF847ZOZ','ENCFF773JYF', 'ENCFF458IQB','ENCFF412EJZ','ENCFF167XAJ','ENCFF360LKQ','ENCFF393ARF','ENCFF678GPZ','ENCFF898HGX','ENCFF276EBZ','ENCFF947JNC','ENCFF505DCA','ENCFF235WMX','ENCFF603QML') and 114 samples of ATAC-seq data [67] (GEO accession GSE86886). Genomic intervals were converted from hg19 (GRCh37) to GRCh38 coordinates using the liftOver tool (https://genome.ucsc.edu/cgi-bin/hgLiftOver). We then used a custom script in R/Bioconductor to determine whether our integration sites overlapped with genomic intervals classified as peaks and hotspots.

## HIV transcriptional profiling

Quantitation of HIV transcripts was performed as previously described [6, 24]. Briefly, nucleic acids were extracted from sorted cells in 1 mL TRI reagent with 2.5 µL polyacryl carrier (both from Molecular Research Center). Total cellular RNA was extracted per the TRI Reagent protocol and total cellular DNA was extracted with 'back extraction buffer' (4M guanidine thiocyanate, 50mM sodium citrate, 1M Tris). A common RT reaction was used to generate cDNA for all ddPCR assays except TAR, for which a separate polyadenylation and reverse transcription was performed. Each 50 µL RT common reaction contained cellular RNA, 5µL of 10x Superscript III buffer (Invitrogen), 5 µL of 50mM $MgCl_2$, 2.5 µL of 50ng/µl random hexamers (Invitrogen), 2.5 µL of 50µM dT15, 2.5 µL of 10mM dNTPs, 1.25 µL of 40U/µL RNaseOUT (Invitrogen), and 2.5 µL of 200U/µL Superscript III RT (Invitrogen). The reverse transcription conditions were as follows: 25.0°C for 10min, 50.0°C for 50min, followed by an inactivation step at 85°C for 5 min. cDNA from each sample was then assayed in duplicate wells for TAR, R-U5-pre-Gag ("Long LTR"), Pol, U3-polyA ("PolyA"), and multiply-spliced Tat-Rev regions

by ddPCR. This panel of assays has been extensively validated in prior studies [6, 24]. Data were normalized to the levels of TERT DNA, which was used as a housekeeping gene. Each 20 µL ddPCR reaction contained 5 µL cDNA or DNA, 10 µL of ddPCR Supermix for Probes (no dUTP) (Bio-Rad), 900 nM of primers, and 250 nM of probe. Following production of droplet emulsions using the QX100 Droplet Generator (Bio-Rad), the samples were amplified under the following thermocycling conditions: 10 minutes at 95˚C, 45 cycles of 30 seconds at 95˚C and 59˚C for 60 seconds, and a final droplet cure step of 10 minutes at 98˚C, using a 7900 Thermal Cycler (Life Technologies). Droplets were quantified using the QX100 Droplet Reader (Bio-Rad) and analyzed using the QuantaSoft software (version 1.6.6, Bio-Rad) in the "Rare Event Detection" quantification mode.

## RNA-Seq

Memory CD4+ T cells from uninfected HLACs with the phenotypes CD57-CD127-, CD57+ CD127-, and CD57-CD127+ were sorted into Trizol LS (Invitrogen) for downstream RNA extraction. Trizol-chloroform phase separation was done per manufacturer's instructions, and RNA was subsequently isolated from the aqueous phase using the RNeasy Micro Kit (Qiagen) by following the RNA cleanup protocol as described in the manufacturer's handbook. RNA-seq libraries were prepared with the ovation RNA-seq system V2 kit (NuGEN). Briefly, total RNA was reverse transcribed to synthesize the first-strand cDNA using a combination of random hexamers and a poly-T chimeric primer. The RNA template was then partially degraded by heating and the second strand cDNA was synthesized using DNA polymerase. The double-stranded DNA was then amplified using single primer isothermal amplification (SPIA). SPIA is a linear cDNA amplification process in which RNase H degrades RNA in DNA/RNA hetero-duplex at the 5′-end of the double-stranded DNA, after which the SPIA primer binds to the cDNA and the polymerase starts replication at the 3′-end of the primer by displacement of the existing forward strand. Random hexamers were then used to amplify the second-strand cDNA linearly. Finally, libraries from the SPIA-amplified cDNA were made using the Ultralow DR library kit (NuGEN). The RNA-seq libraries were analyzed for quality and primer dimers by Bioanalyzer and quantified by QPCR (KAPA) prior to sequencing. High-throughput sequencing was done using a SE (single-end) 50 lane(s) on a HISeq 4000 instrument (Illumina). Fastq files generated using the Illumina pipeline were analyzed using the RNA-seq module of CLC Genomic Workbench 12 (Qiagen). All reads were trimmed based on quality scores of 0.05 using a modified-Mott trimming algorithm which calculates the error probability (Limit-p) for each of the bases in the reads [68, 69], and on the number of ambiguous nucleotides (>2 on ends), and then mapped to the Genome Reference Consortium Human Build 38 reference genome (GRCh38/hg38). This enabled calculating the expression values for every gene in FPKM (Fragment Per Kilobase exon Model per million mapped reads) units [70]. The following default "RNA-seq analysis" settings were used to map reads: minimum length fraction 0.8, minimum similarity fraction 0.8, and maximum number of hits for a read 10. Library size normalization was performed using the TMM (trimmed mean of M values) method [71]. Heatmaps were generated based on hierarchical clustering using the "Create Heat Map" tool in CLC Genomic Workbench. Log CPM (counts per million) values were calculated for each gene, followed by a Z-score normalization across samples for each gene where the counts for each gene were mean centered and scaled to unit variance. The heatmaps of the top 25 genes with the highest coefficients of variation genes across analyzed T cell subsets was calculated based on Euclidean distance. Differentially expressed gene (DEG) lists were generated using the "Differential Expression for RNA-Seq" tool in CLC Genomic Workbench. A statistical differential expression test of the subsets was performed for the set of gene

expression tracks produced from our samples. Each gene was modeled by a separate Generalized Linear Model (GLM) to fit curves to expression values, and the Wald test was used to compare between all pairs of sample groups while fold changes were calculated from the GLM. The filter settings for the DEG lists had a cutoff of a minimum absolute fold change of 1.5, and a False Discovery Rate (FDR) p-value of 0.05. The cut-offs for making the Venn diagram were 1.5 for minimum absolute fold change and 0.05 for maximum FDR p-value.

## Ingenuity Pathway Analysis (IPA)

Lists of differentially expressed transcripts from the 3 comparison populations (CD127+, CD57+, and CD127-CD57- Tm cells) from the 3 independent donors were imported into IPA (Ingenuity Systems, Qiagen). The IPA Core Analysis function was used to calculate directionality (z-score) of up- and down-regulated genes, based on their expression fold-change. The analyzed data sets were then compared using the Comparison Analysis function to interpret the data in the context of biological functions, pathways, and networks. A series of stringent filters was applied, where we limited our sample comparisons to analysis of human pathways → tissue and primary cells → immune cells. Significance of the biological functions and the canonical pathways were tested by the Fisher exact test p-value, and the relevant z-scores, which predict activation or suppression of a given pathway given the differential expression of genes in the comparison, were calculated. For all IPA analyses, an expression p-value cutoff of 0.01 was applied.

For pathway analysis of biological functions, the Canonical Pathways tool of IPA was used to identify pathways that were affected by differentially expressed genes in CD127+ Tm cells versus CD57+ Tm cells, and in CD127+ Tm cells versus CD127-CD57- Tm cells. A heatmap of selected pathways was created, based on a p-value cutoff filter of $\log_{10} 1.3$. Based on canonical pathways of biological functions, two summary graphical representations of selected proteins were created using IPA Path Designer tool. Colors of the pathway components correspond to experimental expression values.

## Supporting information

**S1 Fig. CD127+ Tm cells from tonsils are poorly susceptible to productive infection by HIV-1 even when their frequencies are higher than that of CD57+ Tm cells.** HLACs were exposed for 3 days to the CCR5-tropic reporter virus F4.HSA, after which the CD57+ Tm and CD127+ Tm populations were assessed for infection levels. Shown are results from two independent donors. Note that despite the higher frequencies of CD127+ Tm cells relative to CD57 + Tm cells, the infection rates were markedly lower in the former.
(TIF)

**S2 Fig. Stimulation of HLACs and PBMCs downregulates CD127.** Unstimulated or PHA-stimulated HLACs or PBMCs were mock-treated or infected for 3 days with F4.HSA. Expression levels of CD57 and CD127 were then compared in uninfected memory CD4+ T cells (*blue*) and HIV-infected (HSA+) memory CD4+ T cells (*red*). Results are pre-gated on live, singlet CD3+CD8-CD45RO+CD45RA- cells. The proportion of uninfected and infected cells are indicated in blue and red, respectively.
(TIF)

**S3 Fig. No differential expression of SAMHD1 expression in the tissue Tm subsets.** A) Validation of SAMHD1 antibody by flow cytometry. Unstimulated PBMCs were mock-treated or exposed for 3 days to F4.HSA. Depicted within the gates are the proportions of infected cells amongst the SAMHD1- and SAMHD1+ cells. The results demonstrate that SAMHD1- cells,

which are permissive, harbor a higher proportion of infected cells than non-permissive, SAMHD1+ cells, as expected [3]. Results are pre-gated on live, singlet CD3+CD8-CD45RO+-CD45RA- cells. B) Bar graph of the percentage of SAMHD1+ cells across the three indicated memory CD4+ T cell subsets from HLACs as determined by flow cytometry. Results are pre-gated on live, singlet CD3+CD8-CD45RO+CD45RA- cells. ns: non-significant as determined using a 2-tailed paired parametric t-test. C) SAMHD1 levels in the three Tm subsets from HLACs as determined by Western blot. Beta-actin served as a loading control.
(TIF)

**S4 Fig. Vpx-Vpr does not increase expression of CD57.** HLACs from two donors were sorted for CD127+, CD57+, and CD57-CD127-Tm cells as indicated in the contour dot plots. Cells were then mock-treated or exposed for 3 days to F4.HSA virions harboring Vpx at a ratio of 1:2 or 2:1 as indicated. Three days later, cells were monitored for cell-surface expression levels of CD57 by FACS. The data demonstrate that Vpx did not increase CD57 expression levels in any of the sorted Tm populations.
(TIF)

**S5 Fig. Sorted populations of CD127+, CD57+, and CD57-CD127- Tm cells harbor equivalent levels of mitochondrial DNA.** The indicated populations of cells were sorted from uninfected cultures (*left*) or HIV-infected cultures (*right*). Infected cultures were sorted on HSA-cells to eliminate productively-infected cells from the culture. Each donor specimen is indicated by a different color. n.s.: not significant, as determined by a one-way paired ANOVA using a Bonferroni posttest to account for multiple comparisons.
(TIF)

**S6 Fig. CD127+ Tm cells are an HIV reservoir *in vivo*.** (A) Gut specimens were obtained via sigmoid biopsies from an ART-suppressed, HIV-infected individual treated during chronic infection. The specimens were processed into a single-cell suspension, and sorted for CD127+, CD57+, and CD57-CD127- Tm cells as shown. Because the recovery of CD57+ Tm cells was low, these cells were combined with the CD57-CD127- Tm cells to generate one single population of CD127- Tm cells. (B) The sorted CD127+ and CD127- Tm cells were assayed for the levels of HIV DNA by ddPCR. Data are reported as the HIV DNA copies per microgram of total DNA (*left*), or per million cells as determined by copies of the TERT housekeeping gene (*right*).
(TIF)

**S7 Fig. F4.HSA does not integrate more frequently into non-genic or inaccessible regions of the genome in latently-infected CD127+ Tm cells as compared to latently-infected CD57+ Tm cells.** HSA- CD127+ Tm cells or HSA- CD57+ Tm cells were sorted from F4. HSA-infected cultures from two donors, and assessed for the genomic sites of HIV integration. The frequencies of viral integrations into genic regions of the human genome did not show a consistent trend of in-gene enrichment/depletion between the two subsets (A). Although there was a trend whereby the integration sites in CD127+ Tm cells were less frequently found in accessible regions of the chromosome as compared to in CD57+ Tm cells (as defined by publicly-available DNase-seq and ATAC-seq datasets from CD4+ T cells), this trend was not statistically significant (B). The three data points in each sample correspond to three sampling replicates from the same sorted specimen. Black horizontal lines between dots represent means, and standard deviations are shown as error bars. n.s.: non-significant, as assessed by 2-tailed unpaired student's t-tests.
(TIF)

**S8 Fig. Effects of latency reversal agents and IL7 on latently-infected CD127+ Tm cells.** (A) HLAC cultures were mock-treated or exposed to F4.HSA and then sorted for HSA- CD127 + Tm or naïve CD4+ T cells, which were then cultured with media alone or in the presence of 12 nM of the ingenol PEP005 or 40 nM of the HDAC inhibitor romidepsin. Reactivation levels were compared to stimulation with anti-CD3/CD28. The proportion of infected (HSA+) cells that have downregulated cell-surface CD4 are indicated. Data are representative of one of two donors. (B) HLAC cultures from two donors were either mock-treated or exposed to F4.HSA, and then sorted for HSA- CD127+ Tm cells, which were then stimulated with anti-CD3/CD28 or 10 ng/ml IL7. The top row shows an example of a donor whose latently-infected CD127 + Tm cells reactivated in response to IL7 treatment, while the bottom row shows an example of a different donor whose CD127+ Tm cells did not respond in this manner. The proportions of infected (HSA+) cells that have downregulated cell-surface CD4 are indicated.
(TIF)

**S1 Table. Genes differentially expressed between CD127+ and CD57+ Tm cells.** Only genes with differential expressions of $p < 0.05$ are listed. Genes with positive Log2 fold-change values are upregulated in the CD127+ Tm cells, while those with negative Log2 fold-change values are downregulated in the CD127+ Tm cells. P-values less than $10^{-16}$ are shown as 0.
(XLSX)

**S2 Table. Genes differentially expressed between CD127+ and CD57- CD127- Tm cells.** Only genes with differential expressions of $p < 0.05$ are listed. Genes with positive Log2 fold-change values are upregulated in the CD127+ Tm cells, while those with negative Log2 fold-change values are downregulated in the CD127+ Tm cells. P-values less than $10^{-16}$ are shown as 0.
(XLSX)

**S3 Table. Activation z-scores of IPA pathways identified when comparing CD127+ Tm cells with CD57+ Tm cells, or CD127+ Tm cells with CD57-CD127- Tm cells.** Pathways are listed according to z-score, from top to bottom (highest to lowest absolute z-score) with a p-value cutoff filter of $\log_{10}1.3$. Blue corresponds to pathways downregulated in CD127+ Tm cells, with the intensity of the blue corresponding to the extent of downregulation. Orange corresponds to pathways upregulated in CD127+ Tm cells.
(PDF)

## Acknowledgments

We thank J. McGuire from the Gladstone Genomics Core for assistance in generating the RNAseq libraries and the UCSF Center for Advanced Technology for sequencing; N. Raman for assistance in flow cytometry; N. Manel for the Vpx-Vpr plasmid construct; M. Somsouk for assistance in the gut biopsies; the Massachusetts General Hospital Center for Computational & Integrative Biology DNA Core, specifically N. Stange-Thomann, A. Avery, K. Belanger, and H. Wang for providing the Illumina MiSeq deep sequencing service; G. Maki for assistance in figure preparation; F. Chanut for editorial assistance; and R. Givens for administrative assistance.

## Author Contributions

**Conceptualization:** Guinevere Q. Lee, Marielle Cavrois, Warner C. Greene, Nadia R. Roan.

**Data curation:** Feng Hsiao, Julie Frouard, Sushama Telwatte, Guinevere Q. Lee, Rebecca Hoh, Nadia R. Roan.

**Formal analysis:** Feng Hsiao, Julie Frouard, Andrea Gramatica, Sushama Telwatte, Guinevere Q. Lee, Pavitra Roychoudhury, Xiaoyu Luo, Nadia R. Roan.

**Funding acquisition:** Nadia R. Roan.

**Investigation:** Feng Hsiao, Julie Frouard, Guorui Xie, Sushama Telwatte, Guinevere Q. Lee, Xiaoyu Luo, Nadia R. Roan.

**Methodology:** Feng Hsiao, Julie Frouard, Andrea Gramatica, Guorui Xie, Sushama Telwatte, Guinevere Q. Lee, Pavitra Roychoudhury, Roland Schwarzer, Xiaoyu Luo, Steven A. Yukl, Nadia R. Roan.

**Project administration:** Nadia R. Roan.

**Resources:** Sulggi Lee, Rebecca Hoh, Steven G. Deeks, Nadia R. Roan.

**Supervision:** Steven A. Yukl, R. Brad Jones, Warner C. Greene, Nadia R. Roan.

**Validation:** Nadia R. Roan.

**Visualization:** Nadia R. Roan.

**Writing – original draft:** Nadia R. Roan.

**Writing – review & editing:** Feng Hsiao, Julie Frouard, Andrea Gramatica, Sushama Telwatte, Guinevere Q. Lee, Steven A. Yukl, Sulggi Lee, Marielle Cavrois, Warner C. Greene, Nadia R. Roan.

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
