## [Decision Letter · Decision Letter 0]

12 Nov 2019

Dear Dr. Roan,

Thank you very much for submitting your manuscript "Tissue memory CD4+ T cells expressing IL-7 receptor-alpha (CD127) preferentially support latent HIV-1 infection" (PPATHOGENS-D-19-01972) for review by PLOS Pathogens. Your manuscript was fully evaluated at the editorial level and by independent peer reviewers. The reviewers appreciated the attention to an important problem, but raised some substantial concerns about the manuscript as it currently stands. These issues must be addressed before we would be willing to consider a revised version of your study. We cannot, of course, promise publication at that time.

We therefore ask you to modify the manuscript according to the review recommendations before we can consider your manuscript for acceptance. Your revisions should address the specific points made by each reviewer.

(1) A letter containing a detailed list of your responses to the review comments and a description of the changes you have made in the manuscript. Please note while forming your response, if your article is accepted, you may have the opportunity to make the peer review history publicly available. The record will include editor decision letters (with reviews) and your responses to reviewer comments. If eligible, we will contact you to opt in or out.

(2) Two versions of the manuscript: one with either highlights or tracked changes denoting where the text has been changed; the other a clean version (uploaded as the manuscript file).

Additionally, to enhance the reproducibility of your results, PLOS recommends that you deposit your laboratory protocols in protocols.io, where a protocol can be assigned its own identifier (DOI) such that it can be cited independently in the future. For instructions see http://journals.plos.org/plospathogens/s/submission-guidelines#loc-materials-and-methods

We hope to receive your revised manuscript within 60 days. If you anticipate any delay in its return, we ask that you let us know the expected resubmission date by replying to this email. Revised manuscripts received beyond 60 days may require evaluation and peer review similar to that applied to newly submitted manuscripts.

[LINK]

Sincerely,

Daniel C. Douek

Associate Editor

PLOS Pathogens

Michael Malim

Section Editor

PLOS Pathogens

Kasturi Haldar

Editor-in-Chief

PLOS Pathogens

orcid.org/0000-0001-5065-158X

Grant McFadden

Editor-in-Chief

PLOS Pathogens

orcid.org/0000-0002-2556-3526

Reviewer's Responses to Questions

**Part I - Summary**

Reviewer #1: The present manuscript by Hsiao et al. investigates the mechanisms of HIV latency in three subsets of CD4+ T cells (CD57+, CD127+, and CD57 and CD127 double negative) isolated from human tonsils, and infected in vitro with replication competent HIV. The results presented in the manuscript show that HIV establishes latent infection in CD127+ cells at a deeper level than in the other two subsets. This is not due to pre-integration effects dependent on the restriction factor, SAMHD1, nor is it due to lower integration frequencies. Indeed, this effect is the consequence of lower basal transcription levels in CD127+ cells, which are a cell type with a significantly more quiescent phenotype than the other two subsets.

Since HIV-1 infects primarily residing in lymphoid tissues, the HLAC system is a significant advancement over models that utilize peripheral blood cells. As such, results emerging from this model could provide novel insights into the nature of cells harboring latent HIV, as well as the molecular mechanisms that underlie this phenomenon. While the studies described in the manuscript are intriguing, it still remains unclear whether CD127+ cells represent a significant reservoir of latent HIV infection in vivo. Additional points that must be addressed are indicated below.

Reviewer #2: Hsiao and colleagues report that HIV latency in lymphoid tissues is preferentially established in memory CD4 T cells expressing the IL-7R (CD127) protein. This paper is a follow up to a previous publication of the Roan group, where they used CyTOF to identify HIV-infected cells that had either: HIV entry but did not express an HIV LTR promoter driven reporter, murine HSA (mHSA), to HIV-infected cells that did express mHSA after HIV entry. The surface markers that identified these two HIV-infected states were IL-7R (CD127) and CD57. The manuscript addresses the molecular mechanism for why CD127+ cells allow HIV to enter but not express mHSA, while CD57+ HIV-infected cells complete HIV infection. The conclusions the author make is that lymphoid tissue CD127+ memory CD4+ T cells support HIV latency.

The entire paper's data is from in vitro HIV infections of human lymphocyte aggregate cultures (HLACs) with no validations from HIV+ people samples, which is a major drawback. Especially, since they are eluding to the idea that in vivo an HIV reservoir could be present in CD127+ memory CD4 T cells.

Reviewer #3: The paper from Hsiao et al builds on observations from their previous paper which characterized CD127+ T cells as a memory subset in which, despite HIV entry, the virus was not efficiently expressed in this tissue memory population. New information provided in this paper is that SAMHD1 does not appear to present a block that restricts infection of these cells and that CD127+ cells support HIV-1 replication if activated through CD3+CD28 suggesting that these cells may be important contributors to the latent reservoir. The paper is clearly written and data support the general conclusions. However, the findings seem to incrementally extend the previous 2017 Cell Reports paper and fall short from providing insights into mechanisms that contribute to repression of HIV-1 in the CD127+ cells. Experiments examining multiple latency reversing agents, the role of the microenvornment, a more direct comparison of blood derived CD127+ cells vs HLAC derived cells would allow a better understanding of how HIV is repressed in CD127+ memory cells.

**Part II – Major Issues: Key Experiments Required for Acceptance**

Reviewer #1: All in vitro models are biased one way or another, and the one utilized in this study is no exception. In this case, the bias is that resting (or minimally activated) cells are infected with HIV in the absence of exogenous activation. What is the evidence that in vivo HIV infects resting or minimally activated cells at frequencies comparable to infection of activated cells, thus justifying the extension of these conclusions to the in vivo setting? What is the evidence that this model reflects the in vivo situation more faithfully than model using exogenous stimulation?

Since CD57+, CD127+ and double negative cells present different baseline levels of transcriptional activity (even in the absence of HIV infection) and present different degrees of cell quiescence/activation, it is expected that HIV will integrate with different patterns of genomic distribution in the three subsets. In particular, since CD57+ and double negative cells are more transcriptionally active (i.e. have a much more open chromatin structure) than CD127+ cells, HIV is likely to integrate in euchromatic regions of these subsets more frequently than in CD127+ cells. Have the authors considered whether—and, if so, how—this will influence baseline and re-stimulated HIV expression in the three subsets? Is it possible that in this specific in vitro model, increased frequency of HIV integration in heterochromatic regions also plays a role in the deeper state of latency in CD127+ cells? If so, how does the question above about the physiological relevance of HIV infection of resting (or minimally activated) cells impacts the results?

The role of “chromatin openness” at the time of virus infection and integration is reflected in the evidence that even after re-stimulation with anti-CD3/CD28, the CD127+ subset expresses HIV at significantly lower levels than the other two (see below). While, the studies reported here show that CD127+ cells present much lower levels of basal transcriptional activity (i.e. at the resting state), the studies do not show—nor is it known—whether this is also the case following cell stimulation (i.e. at the activated state). Therefore, the lower levels of HIV expression following re-stimulation (and perhaps to some extent even at baseline in unstimulated cells) may be the consequence of preferential integration in heterochromatic regions of the genome. This ought to be addressed.

Based on the legend to Fig. 5, panel B of that figure shows the copy number of HIV transcripts normalized to total number of CD127+ or total memory cells rather than to the number of infected cells in the total memory and CD127+ memory subsets. Since CD127+ cells contain integrated HIV at frequencies ~2 logs higher than CD57+ and double negative cells (see Fig. 3E), HIV transcription in CD127+ is consequently up to 100-fold (2 logs) lower on a “per infected cell” basis compared to the other two populations. At the same time, following re-stimulation the frequency of HSA+ cells in the CD127+ population is only 6-fold greater than in the total memory population. Again, the CD127+ population contains integrated HIV at frequency 100-fold higher than the other two memory populations, therefore a much smaller fraction of latently infected CD127+ cells can be induced to reactivate HIV compared to CD57+ and double negative cells. In light of these conclusions, and in consideration of the fact that these studies utilized an in vitro model whereby cells are re-stimulated merely 3 days post-infection, it could be argued that CD127+ memory cells are indeed in a state of deep quiescence and the virus they harbor is in a state of deep latency.

How do the authors define the concept of “deep latency”? Since this concept is not quantifiable and is not univocally defined in the literature or in the present manuscript, it is recommended that the authors revise their conclusions in this regard accordingly.

Reviewer #2: There are several interpretations throughout the paper that require clarification and/or additional experiments to demonstrate their overall conclusions that CD127+ memory CD4 T cells would support latently HIV-infected CD4 T cells:

1. In Figure 1 the CD127+ Tmem population is very small in comparison to CD57+ Tmem cells. Since there are not that many CD127+ CD4+ T cells to begin, why isn’t the poor infection due to having fewer of these target cells compared to the CD57+ Tmem cells that are much more abundant? The authors should sort and infect equal number of CD127+ Tmem and CD57+ Tmem cells to address availability of these cells to be infected with HIV.

2. The experiments in Figure 2 and supplemental figure S2 looking at SAMHD1 restriction as a mechanism for blocking mHSA expression does not fully address the authors hypothesis and conclusion. The authors should sort CD127+ Tmem cells, CD57+ Tmem cells, and double negative cells from the HLACs and measure SAMHD1 phosphorylation of threonine 592. Especially since signaling through IL7R induces SAMHD1 phosphorylation of T592. In panel B of figure 2 the data can be interpreted as Vpx-Vpr in HIV-infected CD57 negative cells causes these cells to express CD57, therefore there is an increase in the frequency of CD57+ HIV-infected cells. The authors should address if Vpx-Vpr is activating CD57 expression in contrast to Vpx-Vpr allowing CD57+ Tmem cells to be more permissive to HIV.

3. ddPCR experiments in Figure 3 should normalize the integrated HIV DNA data with a region in the human genome. Normalizing to mitochondria DNA can introduce a source of bias. The authors did not demonstrate CD57+ Tmem. CD127+ Tmem, and double negatives have equal levels of mitochondria DNA, which explains why they chose this DNA for normalization. They should report how many cells were sorted for each population and plot the number of integrated HIV to number of input cells determined by the amplified region of the human genome. A second source of bias is the calculation of integrated HIV DNA to infection rate. The CD127+ Tmem population have a very low frequency of infection therefore a signal of integrated HIV DNA gets amplified when divided to infection frequency.

4. Figure 5 should have experiment that shows the frequency of infection without activation with anti-CD3/CD28 beads. How many cells that are CD127+CD57-CD4+mHSA- spontaneously express mHSA after three days without stimulation? The authors should also perform the same experiment on CD57+ Tmem cells. Will you also get latently HIV-infected cells from CD57+CD127-CD4+mHSA- cells?

Reviewer #3: 1. It is interesting that these cells are maintained in an aggregate culture system. There are opportunities to examine how this microenvironment might influence HIV-1 infection and expression. Experiments separating out cells, determining if there is a role for the IL-7R, adding in blood derived CD127+ cells are some experiments that could maybe address the role of the supporting cells in the establishment of latency.

2. Testing to see if latent infection in CD127+ cells could be induced by different reversing agents would provide insights into if these cells are in general easily reversed or if there maybe blocks to certain pathways. For example, it might be predict that HDACi would have modest effects on inducing HIV in these cells since they are expressing HIV mRNA whereas activating signals cascades upstream of NF-Kb or NFAT, might facilitate transcription and reverse the splicing defect.

3. Was HIV-1 RNA observed in the RNA-seq experiments?

**Part III – Minor Issues: Editorial and Data Presentation Modifications**

Reviewer #1: The quality of the figures is very poor. Although the final figures will have a much greater resolution, these images should have sufficient resolution to be readable by the Reviewers. Some panels cannot be read at all (Fig. 3B-D; Fig. 4B-F; Fig. 5B and 5C). This is not acceptable.

Reviewer #2: There also appears to be a technical flow cytometry staining issue in supplemental figure S2 panel A. The staining does not look to be specific. It is known that SAMHD1 is expressed in every cell, so this data does not add any value to the overall hypothesis of SAMHD1's role in CD127+ Tmem cells.

Reviewer #3: Most of the SAMHD1 related data are presented as supplemental. This finding was presented as an important observation and should be presented in the main text.

PLOS authors have the option to publish the peer review history of their article (what does this mean?). If published, this will include your full peer review and any attached files.

Reviewer #1: No

Reviewer #2: No

Reviewer #3: No

---

## [Decision Letter · Decision Letter 1]

26 Feb 2020

Dear Dr. Roan,

Thank you very much for submitting your manuscript "Tissue memory CD4+ T cells expressing IL-7 receptor-alpha (CD127) preferentially support latent HIV-1 infection" for consideration at PLOS Pathogens. As with all papers reviewed by the journal, your manuscript was reviewed by members of the editorial board and by several independent reviewers. The reviewers appreciated the attention to an important topic. Based on the reviews, we are likely to accept this manuscript for publication, providing that you modify the manuscript according to the review recommendations.

Sincerely,

Daniel C. Douek

Associate Editor

PLOS Pathogens

Michael Malim

Section Editor

PLOS Pathogens

Kasturi Haldar

Editor-in-Chief

PLOS Pathogens

orcid.org/0000-0001-5065-158X

Michael Malim

Editor-in-Chief

PLOS Pathogens

orcid.org/0000-0002-7699-2064

Reviewer Comments (if any, and for reference):

Reviewer's Responses to Questions

**Part I - Summary**

Reviewer #1: (No Response)

Reviewer #2: Hsiao and colleagues have improved their manuscript since the original submission. They did address the major concerns that I brought up. However, based on the new data there are still are major and minor issues that need to be addressed.

Reviewer #3: In general, this is an interesting paper that identifies CD127+ Tm cells as a potential contributor to the HIV reservoir. The paper also highlights the important role that tissue environment may play in shaping T cell function and HIV replication. The paper does begin to address mechanism that maintains and establishes latency in this T cell subset. Overall, the authors were responsive to the original reviews and added requested supporting data. There is also an appreciation that these are difficult experiments using limiting cell numbers. Some minor points, Fig. 3 and its description is still somewhat confusing; the rationale for mt DNA vs HIV copy per cell was not clear, it should be stated how values in 3E were normalized and finally the labeling of 3C top panel is incorrect.

**Part II – Major Issues: Key Experiments Required for Acceptance**

Reviewer #1: (No Response)

Reviewer #2: Major issues:

1. The experiments of supplemental figure 7 can be misleading. DNAseq and ATACseq experiments are biased towards detecting open DNA in promoters and enhancers. There are no reports that HIV integrates into these genomic elements. The data presented cannot be interpreted as “HIV more frequently integrated into closed, inaccessible regions…” and then end with “they [CD127+ cells] do not markedly favor integration of HIV into closed regions of the genome.” This result section has to be re-interpreted and re-written. The discussion portion of this conclusion also has to be re-written to address what is known about the epigenetic makeup of open/closed promoters and the corresponding adjacent gene bodies.

2. Figure 5 panel C shows higher or equal level of productive infection without stimulation in the CD127+ cells in comparison to CD57+ cells. I thought the CD127+ cells were not becoming productively infected but instead undergoing latency. In addition, the stimulation of CD127+ and CD57+ cells are resulting the same level of productive infection, there is an 11-fold increase HIV infection from unstimulated to stimulated. Is this data representative?

3. The authors never address the intactness or replication-competency of the HIV provirus in either CD127+ or CD127- memory CD4 T cells in their in vitro work. They should mention that their findings have a limitation in terms of understanding if there are plenty of intact replication-competent proviruses in CD127+ or CD127- cells to merit these populations harboring the latent replication-competent HIV reservoir in cART.

Reviewer #3: No major issues

**Part III – Minor Issues: Editorial and Data Presentation Modifications**

Reviewer #1: (No Response)

Reviewer #2: Minor issues:

1. Figure 3, panel C the top flow plot shows the three populations that are being sorted, however the x- and y-labels are incorrect.

2. The primary conclusion of Figure 3 is that CD127+ memory CD4 T cells support latent HIV infection, because there are higher levels of integrated HIV copies in the CD127+ cells when compared to CD57+ or CD57-CD127- cell. However, Figure 2, the CD57+ and CD57-CD127- cells have a higher level of productive infection that suggests the productive infection is from non-integrated HIV proviruses since you are showing very little integrated HIV DNA in these cells. These results have to be better explained.

3. The conclusion from the data showing SAMHD1 does not impact productive infection of CD127+ cells doesn’t fully address the experiments addressing the role of SAMHD1 restricting HIV infection. Did the Vpx-Vpr infections of CD127+ cells lead to an increase of integrated HIV DNA in these cells?

4. Their gut biopsy of ART-suppressed HIV-infected individual CD127+ and CD127- memory CD4 T cells had detectable HIV DNA. That mean that both populations can potentially support HIV latency under cART, therefore this observation needs to be addressed equally, instead of bias their conclusion toward the CD127+ cells. Limitations should be addressed in regard to this findings and the in vitro data.

Reviewer #3: Need to provide a better rationale for controls to normalize DNA levels in fig 3.

Figure 3 labeling needs to be corrected

PLOS authors have the option to publish the peer review history of their article (what does this mean?). If published, this will include your full peer review and any attached files.

Reviewer #1: No

Reviewer #2: No

Reviewer #3: No
---

## [Editor Report · Decision Letter 2]

3 Mar 2020

Dear Dr. Roan,

We are pleased to inform you that your manuscript 'Tissue memory CD4+ T cells expressing IL-7 receptor-alpha (CD127) preferentially support latent HIV-1 infection' has been provisionally accepted for publication in PLOS Pathogens.

Best regards,

Daniel C. Douek

Associate Editor

PLOS Pathogens

Michael Malim

Section Editor

PLOS Pathogens

Kasturi Haldar

Editor-in-Chief

PLOS Pathogens

orcid.org/0000-0001-5065-158X

Michael Malim

Editor-in-Chief

PLOS Pathogens

orcid.org/0000-0002-7699-2064
---

## [Editor Report · Acceptance letter]

26 Mar 2020

Dear Dr. Roan,

We are delighted to inform you that your manuscript, "Tissue memory CD4+ T cells expressing IL-7 receptor-alpha (CD127) preferentially support latent HIV-1 infection," has been formally accepted for publication in PLOS Pathogens.

Best regards,

Kasturi Haldar

Editor-in-Chief

PLOS Pathogens

orcid.org/0000-0001-5065-158X

Michael Malim

Editor-in-Chief

PLOS Pathogens

orcid.org/0000-0002-7699-2064